# Zika virus remodels and hijacks IGF2BP2 ribonucleoprotein complex to promote viral replication organelle biogenesis

Clément Mazeaud[1], Stefan Pfister[2], Jonathan E Owen[3], Higor Sette Pereira[4], Flavie Charbonneau[1], Zachary E Robinson[4], Anaïs Anton[1], Cheyanne L Bemis[3], Aïssatou Aïcha Sow[1], Trushar R Patel[4], Christopher J Neufeldt[3], Pietro Scaturro[2], Laurent Chatel-Chaix[1,5,6,7]*

[1]Centre Armand-Frappier Santé Biotechnologie, Institut National de la Recherche Scientifique, Laval, Canada; [2]Leibniz Institute of Virology, Hamburg, Germany; [3]Department of Microbiology and Immunology, Emory University School of Medicine, Atlanta, United States; [4]Department of Chemistry and Biochemistry, Alberta RNA Research and Training Institute, University of Lethbridge, Lethbridge, Canada; [5]Center of Excellence in Research on Orphan Diseases-Fondation Courtois, Quebec, Canada; [6]Regroupement Intersectoriel de Recherche en Santé de l'Université du Québec, Quebec, Canada; [7]Swine and Poultry Infectious Diseases Research Centre, Quebec, Canada

*For correspondence:
laurent.chatel-chaix@inrs.ca

Competing interest: The authors declare that no competing interests exist.

## eLife assessment

This **valuable** study combines multidisciplinary approaches to examine the role of insulin-like growth factor 2 mRNA-binding protein 2 (IGF2BP2) as a potential novel host dependency factor for Zika virus. The main claims are supported by the data but remain **incomplete**. The evidence would be strengthened by improving the western blot analyses and adjusting the toning of their claims in relation to the role of IGF2BP2 for viral replication. With the experimental evidence strengthened, this work will be of interest to virologists working on flaviviruses.

**Abstract** Zika virus (ZIKV) infection causes significant human disease that, with no approved treatment or vaccine, constitutes a major public health concern. Its life cycle entirely relies on the cytoplasmic fate of the viral RNA genome (vRNA) through a fine-tuned equilibrium between vRNA translation, replication, and packaging into new virions, all within virus-induced replication organelles (vROs). In this study, with an RNA interference (RNAi) mini-screening and subsequent functional characterization, we have identified insulin-like growth factor 2 mRNA-binding protein 2 (IGF2BP2) as a new host dependency factor that regulates vRNA synthesis. In infected cells, IGF2BP2 associates with viral NS5 polymerase and redistributes to the perinuclear viral replication compartment. Combined fluorescence *in situ* hybridization-based confocal imaging, *in vitro* binding assays, and immunoprecipitation coupled to RT-qPCR showed that IGF2BP2 directly interacts with ZIKV vRNA 3' nontranslated region. Using ZIKV sub-genomic replicons and a replication-independent vRO induction system, we demonstrated that IGF2BP2 knockdown impairs *de novo* vRO biogenesis and, consistently, vRNA synthesis. Finally, the analysis of immunopurified IGF2BP2 complex using quantitative mass spectrometry and RT-qPCR revealed that ZIKV infection alters the protein and RNA interactomes of IGF2BP2. Altogether, our data support that ZIKV hijacks and remodels the IGF2BP2 ribonucleoprotein complex to regulate vRO biogenesis and vRNA neosynthesis.

## Introduction

Zika virus (ZIKV) is the causative agent in several epidemics in the last 10 years, with the biggest outbreak in Latin America in 2015, making ZIKV a major public health concern. ZIKV belongs to the *Orthoflavivirus* genus within the *Flaviviridae* family which comprises over 70 arthropod-borne viruses (arboviruses) such as dengue virus (DENV), West Nile virus (WNV), and yellow fever virus. Similar to the closely related DENV, ZIKV is primarily transmitted to humans by *Aedes* spp. mosquito bites (*Gould and Solomon, 2008*) and can cause Guillain-Barré syndrome in adults. When vertically transmitted to a fetus during pregnancy, this neurotropic virus can infect the placenta as well as neural progenitor cells, microglial cells, and astrocytes in the developing brain, which may eventually cause severe microcephaly in newborns (*Cugola et al., 2016*; *Krauer et al., 2017*; *Nowakowski et al., 2016*; *Pierson and Graham, 2016*; *Retallack et al., 2016*; *Vota et al., 2021*). Unfortunately, no antiviral treatments against ZIKV or prophylactic vaccines are currently approved. It is thus important to better understand the molecular mechanisms controlling the ZIKV replication in order to identify new therapeutic targets.

ZIKV is a single-stranded positive sense enveloped RNA virus. Its ~11-kb-long viral RNA (vRNA) genome is composed of an open reading frame flanked by 5' and 3' nontranslated regions (NTRs). The vRNA plays a central role in different steps of the viral replication cycle including being translated to produce all viral proteins, used as template for genome neosynthesis, and selectively packaged into assembling viral particles. The vRNA, translated at the endoplasmic reticulum (ER) membrane, encodes one polyprotein which is matured into 10 viral proteins. Three structural proteins (C, prM, and E) together with vRNA assemble new viral particles, and seven nonstructural proteins (NS1, NS2A, NS2B, NS3, NS4A, NS4B, and NS5) are responsible for vRNA neosynthesis, via the activity of the RNA-dependent RNA polymerase (RdRp) NS5 along with the contribution of other viral proteins and host factors (*Acosta et al., 2014*; *Mazeaud et al., 2018*). To efficiently replicate, orthoflaviviruses have evolved to establish a fine-tuned equilibrium between vRNA translation, replication, and encapsidation steps since they cannot occur simultaneously. However, the mechanisms underlying the spatio-temporal fate of vRNA remain elusive. These processes occur in viral replication organelles (vRO) which result from major alterations of the ER and are believed to coordinate in time and space the steps of the life cycle, notably through the spatial segregation of the vRNA, at different steps of the life cycle. vROs accumulate in a large cage-like compartment which is located in the perinuclear area (*Cortese et al., 2017*). Three architectures of these viral organelles-like ultrastructures have been described: vesicle packets (VP), virus bags (VB), and convoluted membranes (CM). The VPs resulting from ER invagination are spherical vesicles with a diameter of approximately 90 nm connected to the cytosol through a 10-nm-wide pore-like opening (*Cortese et al., 2017*; *Hamel et al., 2015*; *Paul and Bartenschlager, 2013*; *Welsch et al., 2009*). They are believed to host vRNA synthesis. Indeed, the nonstructural proteins NS5, NS3, NS1, NS4A, and NS4B, absolutely required for vRNA replication, and also double-stranded RNA (dsRNA), the vRNA replication intermediate, are enriched in VPs as imaged by immunogold labeling followed by transmission electron microscopy (*Chatel-Chaix et al., 2016*; *Welsch et al., 2009*). It has been proposed that the RNA genome exit VPs through their pore to be directly packaged into assembling particles budding into the ER lumen. Immature virions geometrically accumulate in dilated ER cisternae referred to as VBs. The viral and cellular determinants of vRO biogenesis are poorly understood mostly because this process will be impacted (even indirectly) by any perturbations of viral replication efficiency and hence, viral factors abundance. The recent advent of replication-independent VP induction systems identified several determinants of *de novo* VP biogenesis, most notably including ZIKV NS4A, NS1, and 3' NTR RNA, as well as host factors ATL2, VCP, and RACK1 (*Cerikan et al., 2020*; *Cortese et al., 2021*; *Goellner et al., 2020*; *Mazeaud et al., 2021*; *Neufeldt et al., 2019*; *Płaszczyca et al., 2019*; *Shue et al., 2021*).

Multiple reports have shown that cellular RNA-binding proteins (RBPs) regulate the replication, translation, and/or encapsidation of vRNA often through associations with 5' or 3' NTRs (for review, see *Mazeaud et al., 2018*). For instance, YBX1 associates with 3' NTR of DENV vRNA and regulates both vRNA translation and viral particle production (*Diosa-Toro et al., 2022*; *Paranjape and Harris, 2007*; *Phillips et al., 2016*). However, it remains unclear whether all these proteins are co-opted and regulate replication within a single ribonucleoprotein (RNP) complex. In case of the hepatitis C virus (HCV), a *Flaviviridae* from the *Hepacivirus* genus, it was shown that the YBX1 RNP comprising HCV vRNA, IGF2BP2, LARP1, DDX6, and other cell proteins regulates the equilibrium between vRNA

replication and infectious particle production (*Chatel-Chaix et al., 2013*; *Chatel-Chaix et al., 2011*). However, it is unknown whether this is a conserved co-opting mechanism in the *Orthoflavivirus* genus (which also belongs to the *Flaviviridae* family) and if so, whether these viruses modulate the composition of this complex to rewire its functions in favor of replication. Finally, considering that the 3' NTR of vRNA is a key player of vRO morphogenesis (*Cerikan et al., 2020*), this raises the hypothesis that this viral process involves host vRNA-binding proteins prior to vRNA synthesis.

To determine whether the co-opting of these RNA-binding proteins was conserved in the *Orthoflavivirus* genus, we have assessed in this study the potential role of 10 host RBPs in ZIKV and DENV replication. We have identified insulin-like growth factor 2 mRNA-binding protein 2 (IGF2BP2) as a new host dependency factor for the ZIKV life cycle which regulates vRNA synthesis. In infected cells, IGF2BP2 associates with NS5 viral polymerase and vRNA 3' NTR. Importantly, IGF2BP2 impairs *de novo* vRO biogenesis and consistently, vRNA synthesis. Finally, ZIKV infection alters the protein and RNA interactomes of IGF2BP2. Altogether, our data support that ZIKV hijacks and remodels the IGF2BP2 ribonucleoprotein complex, to regulate vRO biogenesis and vRNA neosynthesis.

## Results

### IGF2BP2 regulates ZIKV replication cycle

To identify new cellular proteins regulating ZIKV and DENV vRNA functions during viral replication, we hypothesized that these orthoflaviviruses have evolved to share conserved host co-opting mechanisms with HCV which belongs to the *Hepacivirus* genus within the *Flaviviridae* family like the *Orthoflavivirus* genus. Based on our previous work with host RBPs regulating HCV life cycle (*Chatel-Chaix et al., 2013*; *Chatel-Chaix et al., 2011*), we performed a targeted small-scale RNA interference (RNAi) screening assessing the requirement of 10 host RBPs for ZIKV and DENV replication. Protein knockdown (KD) in human hepatocarcinoma Huh7.5 cells was achieved through the transduction of short-hairpin RNA (shRNA)-expressing lentiviruses, whose respective efficiencies in this cell line were previously validated (*Chatel-Chaix et al., 2013*; *Chatel-Chaix et al., 2011*). None of these shRNA had any impact on cell viability as measured by 3-(4,5-dimethylthiazol-2-yl)-2,5-diphenyltetrazolium bromide (MTT) assays (*Figure 1A*). Two days post-transduction, cells were infected with pathogenic contemporary ZIKV H/PF/2013 strain (Asian lineage), ancestral ZIKV MR766 strain (African lineage), or serotype 2 DENV2 16681s strain. Two days post-infection (dpi), infectious viral particle production was measured by plaque assays (*Figure 1B–D*). The KD of DEAD-box helicases RHA (RNA helicase A, also named DHX9), DDX6, DDX21, or DDX5 had no significant impact in virus production when compared to the non-target control shRNA (shNT). In contrast, the KD of IGF2BP2 and LARP1 resulted in significant decrease and increase in the titers of both ZIKV strains, respectively. The observed phenotypes were specific to ZIKV since no significant impact was observed for DENV (*Figure 1B–D*).

To validate these phenotypes, we took advantage of ZIKV FSS13025 strain-based reporter viruses expressing Renilla luciferase (Rluc), whose activity in infected cells can be used as a read-out of overall vRNA replication (*Figure 1E*, *Fischl and Bartenschlager, 2013*; *Shan et al., 2016*). IGF2BP2 KD resulted in a significant 60% decrease in viral replication. In contrast, LARP1 KD had no stimulatory effect on viral replication, suggesting a role of this host factor in virus assembly and/or release. Of note YBX1, YBX2, and DDX3 KD all resulted in a decrease in ZIKV replication although this did not translate into impacts in virus titers (*Figure 1B–F*). We decided to focus further investigation on IGF2BP2 since it was our best candidate in terms of ZIKV replication impairment upon KD (*Figure 1B and C*). First, western blotting and RT-qPCR with transduced Huh7.5 cell lysates validated that IGF2BP2 was efficiently knocked down upon transduction at both protein and mRNA levels (*Figure 2A and B*), which correlated with a 74% decrease in ZIKV production (*Figure 2C*) and reductions in the expression of viral proteins NS3, NS4A, and NS5 (*Figure 2A*). Comparable impairment of ZIKV titers upon IGF2BP2 KD was observed in other cell lines relevant for ZIKV pathogenesis, namely human immortalized astrocytes (NHA-hTERT; *Figure 2D*) and cancer-derived trophoblasts (JEG-3; *Figure 2E*) as well as for a third wild-type ZIKV strain in Huh7.5 cells (*Figure 3A*), i.e., ZIKV FSS13025 (Asian lineage, isolated in Cambodia in 2010).

Our initial screen showed that IGF2BP2 KD had minimal impact on DENV2 replication, suggesting a ZIKV-specific phenotype (*Figure 1D*). To confirm this, we have included in our workflow four additional strains of DENV representing the four known serotypes. IGF2BP2 KD did not change the production

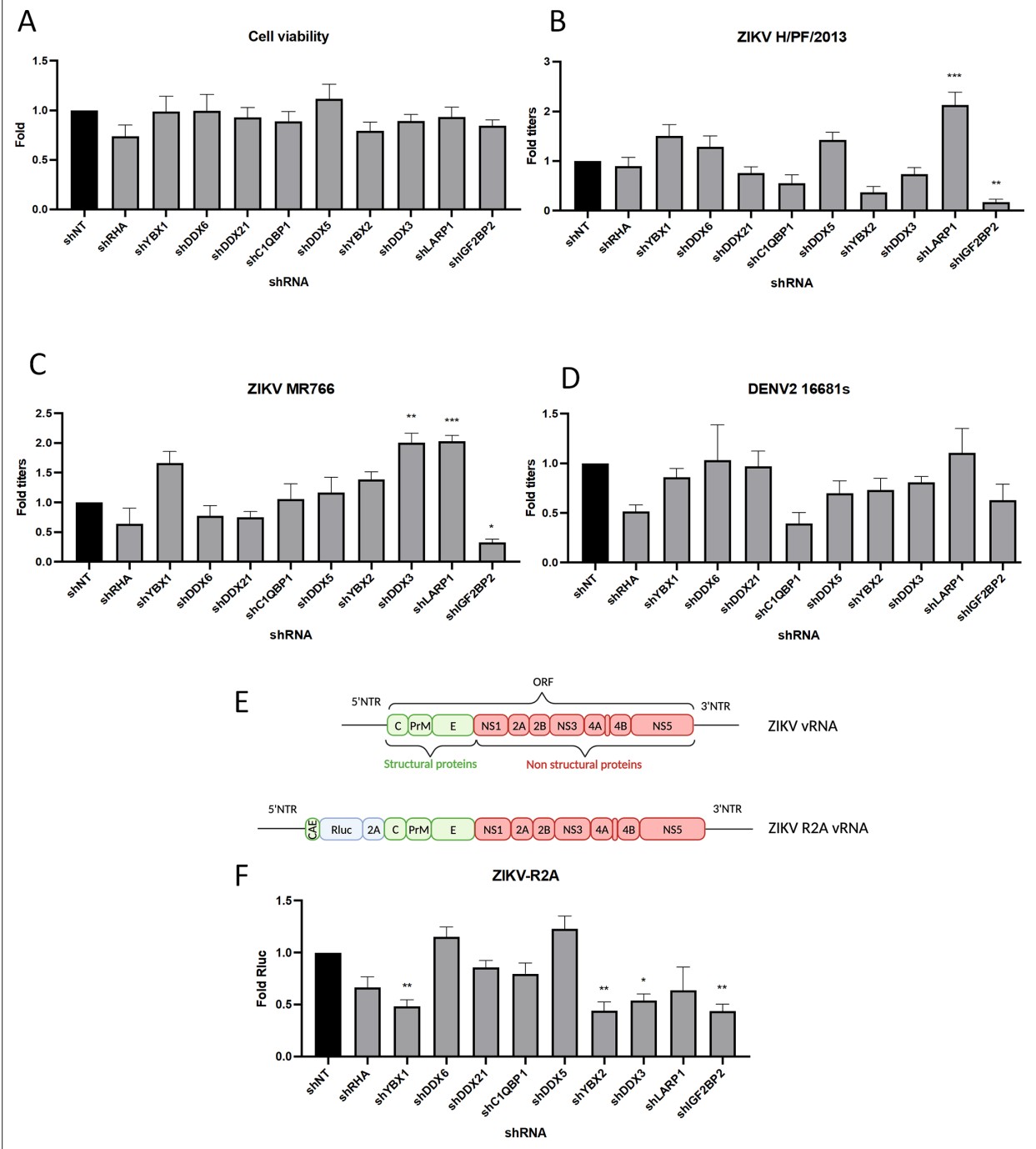

**Figure 1.** A RNA interference (RNAi) mini-screening of RNA-binding proteins to identify host factors involved in dengue virus (DENV) and Zika virus (ZIKV) replications. Huh7.5 were transduced with short-hairpin RNA (shRNA)-expressing lentiviruses at an MOI of 5–10. (**A**) Four days post-transduction, 3-(4,5-dimethylthiazol-2-yl)-2,5-diphenyltetrazolium bromide (MTT) assays were performed to evaluate cytotoxicity effect of the knockdown (KD). Two days post-transduction cells were infected with either (**B**) ZIKV H/PF/2013, (**C**) ZIKV MR766, or (**D**) DENV2 16681s at an MOI of 0.01. 48 hr post-infection, the production of infectious viral particles was evaluated by plaque assays. (**E**) Schematic of the Renilla luciferase (Rluc)-expressing ZIKV reporter virus (ZIKV-R2A) based on the FSS13025 isolate (Asian lineage). (**F**) Cells were prepared, exactly as in B–D but infected with ZIKV-R2A at an MOI of 0.001. 48 hr post-infection, cells were lysed and bioluminescence was measured and normalized to the control cells expressing a non-target shRNA (shNT). Means ± SEM are shown based on three to five independent experiments for each shRNA. p<0.0001; ***: p<0.001; **: p<0.01; *p<0.05 (one-way ANOVA test).

The online version of this article includes the following source data for figure 1:

**Source data 1.** Data points to generate all bar graphs of *Figure 1*.

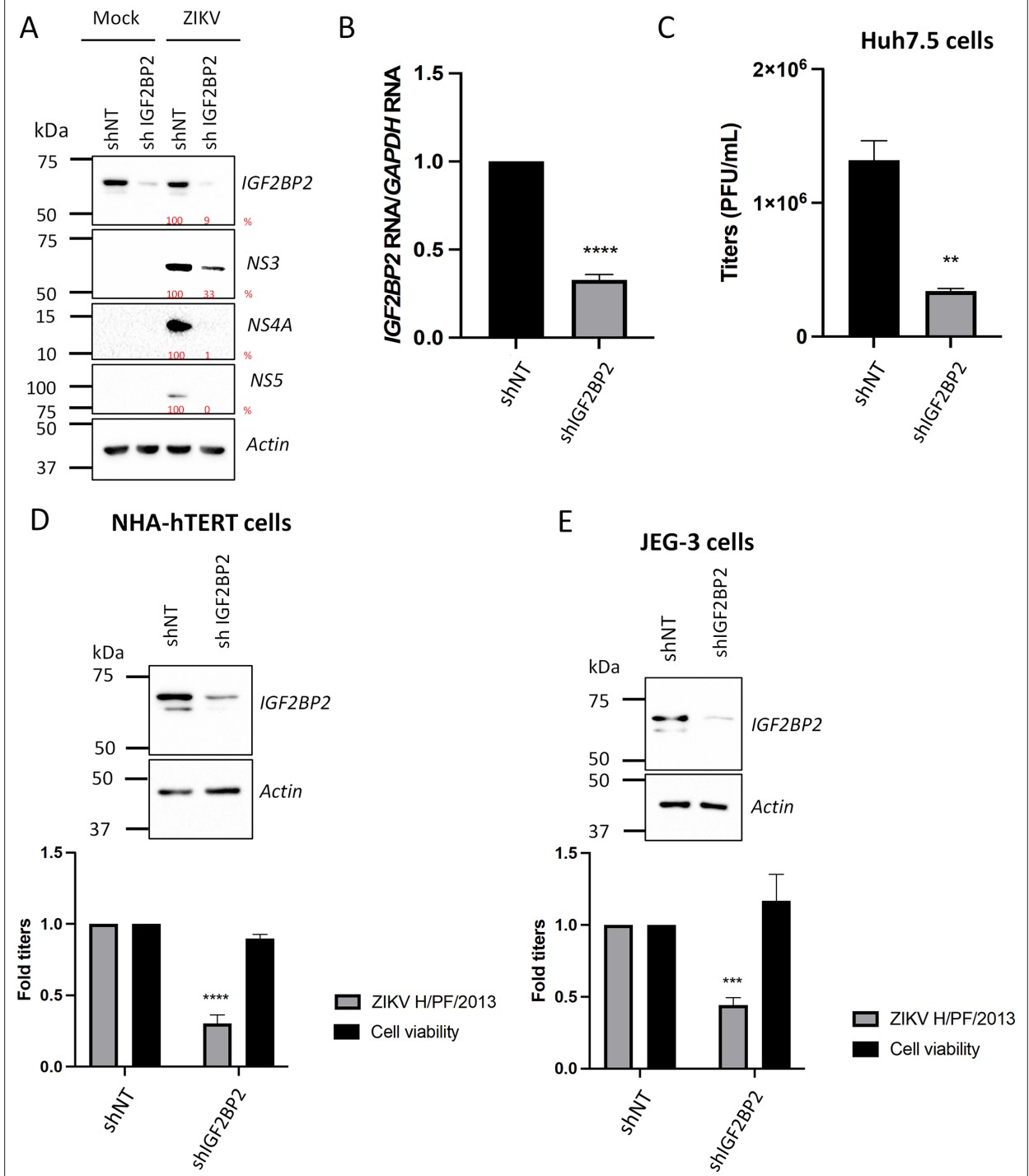

**Figure 2.** IGF2BP2 positively regulates Zika virus (ZIKV) replication in multiple cell lines. Liver Huh7.5 (**A–C**), astrocytic NHA-hTERT (**D**), and placental JEG-3 (**E**) cells were transduced with non-target shRNA (shNT) or shIGF2BP2 lentiviruses at an MOI of 10. Two days post-transduction, cells were infected with ZIKV H/PF/2013 at an MOI between 0.01 and 1 depending on the cell line. Two days post-infection, supernatant and cells were collected. IGF2BP2 expression at the protein level (A, D, E; all cell lines) and mRNA level (B; Huh7.5 cells) were evaluated by western blotting (WB) and RT-qPCR, respectively. Cell supernatants were used for plaque assays (**C–E**). For NHA-hTERT and JEG-3, the supernatant and cells are collected for titration and WB, respectively (**E–F**). 3-(4,5-Dimethylthiazol-2-yl)-2,5-diphenyltetrazolium bromide (MTT) assays were performed to assess the cell viability in transduced NHA-hTERT and JEG-3 cells (**D–E**). Means ± SEM are shown based on five (**D**), three (**C**), and four (**D–E**) independent experiments. ****: $p < 0.0001$; ***: $p < 0.001$; **: $p < 0.01$ (unpaired t-test).

The online version of this article includes the following source data for figure 2:

*Figure 2 continued on next page*

*Figure 2 continued*

**Source data 1.** Data points to generate the bar graphs of *Figure 2B–E*.

**Source data 2.** PDF file containing original western blots for *Figure 2A and D–E*, indicating the relevant bands and conditions.

**Source data 3.** Original files for western blot analysis displayed in *Figure 2A and D–E*.

of DENV1, DENV2, or DENV4, while DENV3 replication was only slightly decreased (*Figure 3B–E*). As control, treatment with NITD008, an inhibitor of flaviviral NS5 polymerase (*Deng et al., 2016*; *Lo et al., 2016*; *Nelson et al., 2015*; *Yin et al., 2009*), completely abrogated viral particle production for all orthoflaviviruses tested. Finally, IGF2BP2 KD had no impact on the replication of either WNV, another orthoflavivirus (*Figure 3F*), or of SARS-CoV-2, another positive-strand RNA virus from the *Coronaviviru* family (*Figure 3G*). Overall, these data clearly show that the IGF2BP2 is a specific host dependency factor for ZIKV replication.

## IGF2BP2 redistributes to ZIKV replication complexes in infected cells

Following the observations that IGF2BP2 is a ZIKV host dependency factor, we investigated the impact of ZIKV infection on IGF2BP2 expression and subcellular distribution. We performed western blotting on extracts of either uninfected or ZIKV-infected Huh7.5 cells. IGF2BP2 expression remained unchanged at 2 and 3 dpi in contrast to DDX3 and DDX5 whose levels were decreased over the course of infection (*Figure 4A*). Then, we investigated the localization of IGF2BP2 in infected cells using confocal microscopy. In uninfected cell, IGF2BP2 exhibited a homogenous distribution throughout the cytoplasm. In ZIKV-infected cells, an accumulation of IGF2BP2 in areas enriched for NS3 and viral dsRNA was observed at 2 dpi (*Figure 4B*), correlating with a partial colocalization with NS3 and dsRNA (Manders' coefficients of 0.45±0.04 and 0.46±0.04, respectively). dsRNA is an intermediate of vRNA replication and hence, anti-dsRNA antibodies detect replication complexes, which along with VPs, typically accumulate in a cage-like region in the perinuclear area with viral proteins required for vRNA replication step, such as NS3 and NS5 (*Cortese et al., 2017*). This compartment is typically surrounded by NS3-positive CMs which are devoid of dsRNA (*Anton et al., 2021*; *Chatel-Chaix et al., 2016*; *Cortese et al., 2017*). A similar, yet less pronounced, phenotype was observed at an earlier time point (1 dpi, *Figure 4—figure supplement 1*).

The relocalization of IGF2BP2 into the replication compartment raised the hypothesis that this protein interacts with viral proteins involved in vRNA replication. To test this, we generated Huh7.5 cells stably expressing HA-tagged IGF2BP2 to subsequently perform co-immunoprecipitation assays (*Figure 4C*). As control, we first confirmed that this tagged recombinant protein, similarly to endogenous IGF2BP2, also redistributed to the dsRNA-positive area in ZIKV-infected cells (*Figure 4—figure supplement 2*). We next performed anti-HA immunoprecipitations with extracts from uninfected and ZIKV-infected cells. Western blot analysis of immunopurified complexes showed that IGF2BP2-HA specifically associates with ZIKV NS5 polymerase but not with NS3 protease/helicase (*Figure 4C and D*). Overall, these data show that ZIKV infection induces the physical recruitment of IGF2BP2 to the replication compartment, and further suggest that IGF2BP2 might contribute to vRNA synthesis through interactions with NS5 polymerase.

## IGF2BP2 associates with ZIKV RNA

Considering that IGF2BP2 associates with NS5 in the replication compartment and that it possesses RNA-binding activities, we evaluated whether IGF2BP2 interacts with ZIKV vRNA. IGF2BP2 along with IGF2BP1 and IGF2BP3 paralogues are highly conserved oncofetal RBPs belonging to the insulin-like growth factor 2 mRNA-binding protein family, which typically regulate the post-transcriptional regulation of cellular RNAs. With their roles in the splicing, transport, translation, and stabilization of a wide variety of RNAs (*Bell et al., 2013*; *Degrauwe et al., 2016*), IGF2BP1, 2, and 3 are involved in numerous cellular function, such as differentiation, migration, metabolism, and proliferation (*Bell et al., 2013*; *Degrauwe et al., 2016*). IGF2BP2 is composed of six canonical RNA-binding domains, namely two N-terminal RNA recognition motifs (RRM) and four C-terminal human heterogeneous nuclear ribonucleoprotein-K homology (KH) domains (*Nielsen et al., 1999*; *Nielsen et al., 2004*; *Wächter et al., 2013*).

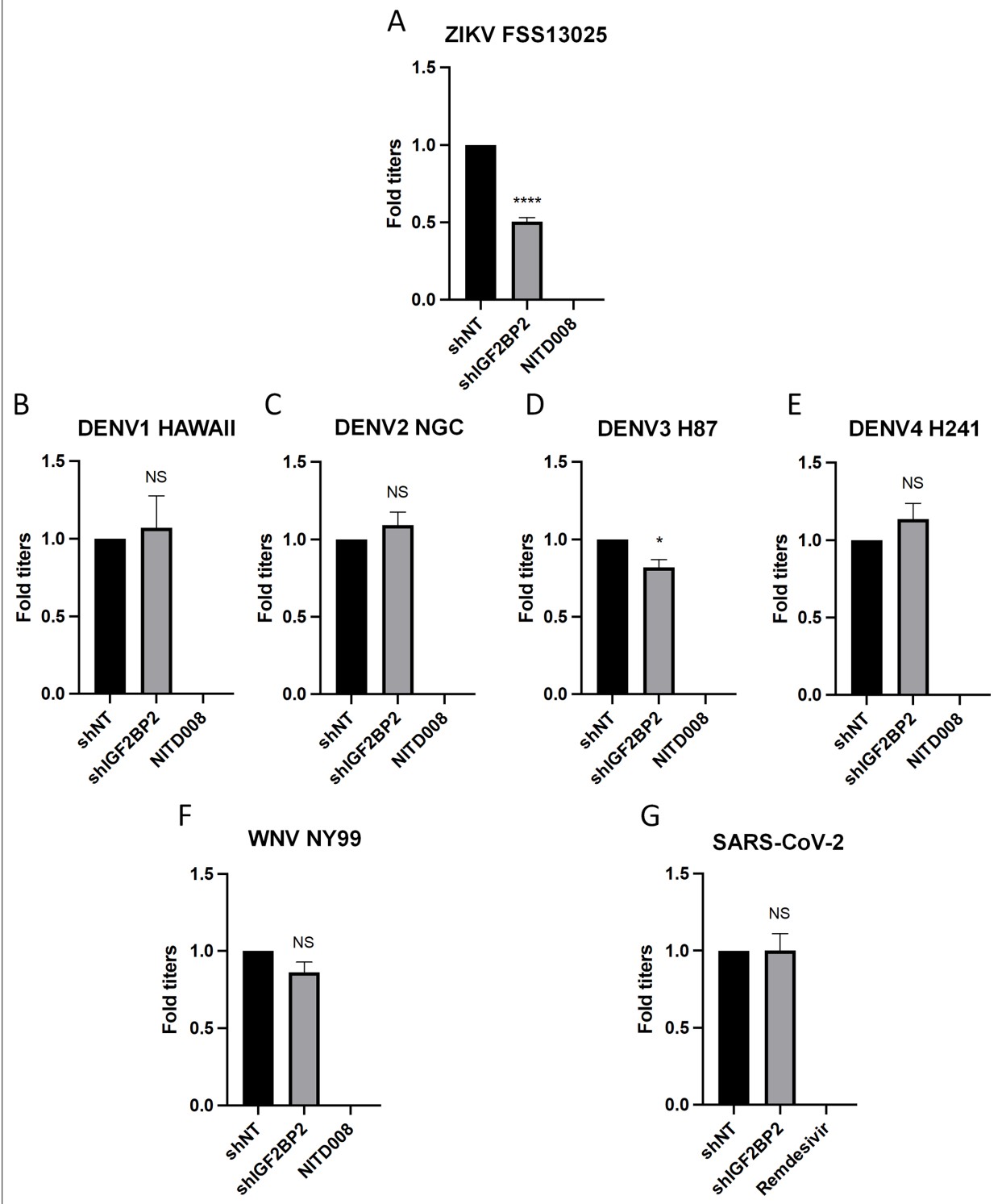

**Figure 3.** IGF2BP2 dependency is Zika virus (ZIKV)-specific. Huh7.5 cells were transduced with non-target shRNA (shNT) or shIGF2BP2 lentiviruses at an MOI of 10. Two days post-transduction cells were infected with (**A**) ZIKV FSS13025, (**B**) dengue virus (DENV)1 HAWAII, (**C**) DENV2 NGC, (**D**) DENV3 H87, (**E**) DENV4 H241, (**F**) West Nile virus (WNV) NY99, (**G**) SARS-CoV-2 at an MOI of 0.1. Virus-containing cell supernatants were collected and titrated 2 days post-infection by plaque assays. Treatment with RNA-dependent RNA polymerase (RdRp) inhibitors NITD008 and Remdesivir were used as positive controls of replication inhibition of orthoflaviviruses and SARS-CoV-2, respectively. Means ± SEM are shown based on three independent experiments. ****: $p<0.0001$; *: $p<0.05$; NS: not significant (unpaired t-test).

The online version of this article includes the following source data for figure 3:

**Source data 1.** Data points to generate all bar graphs of *Figure 3*.

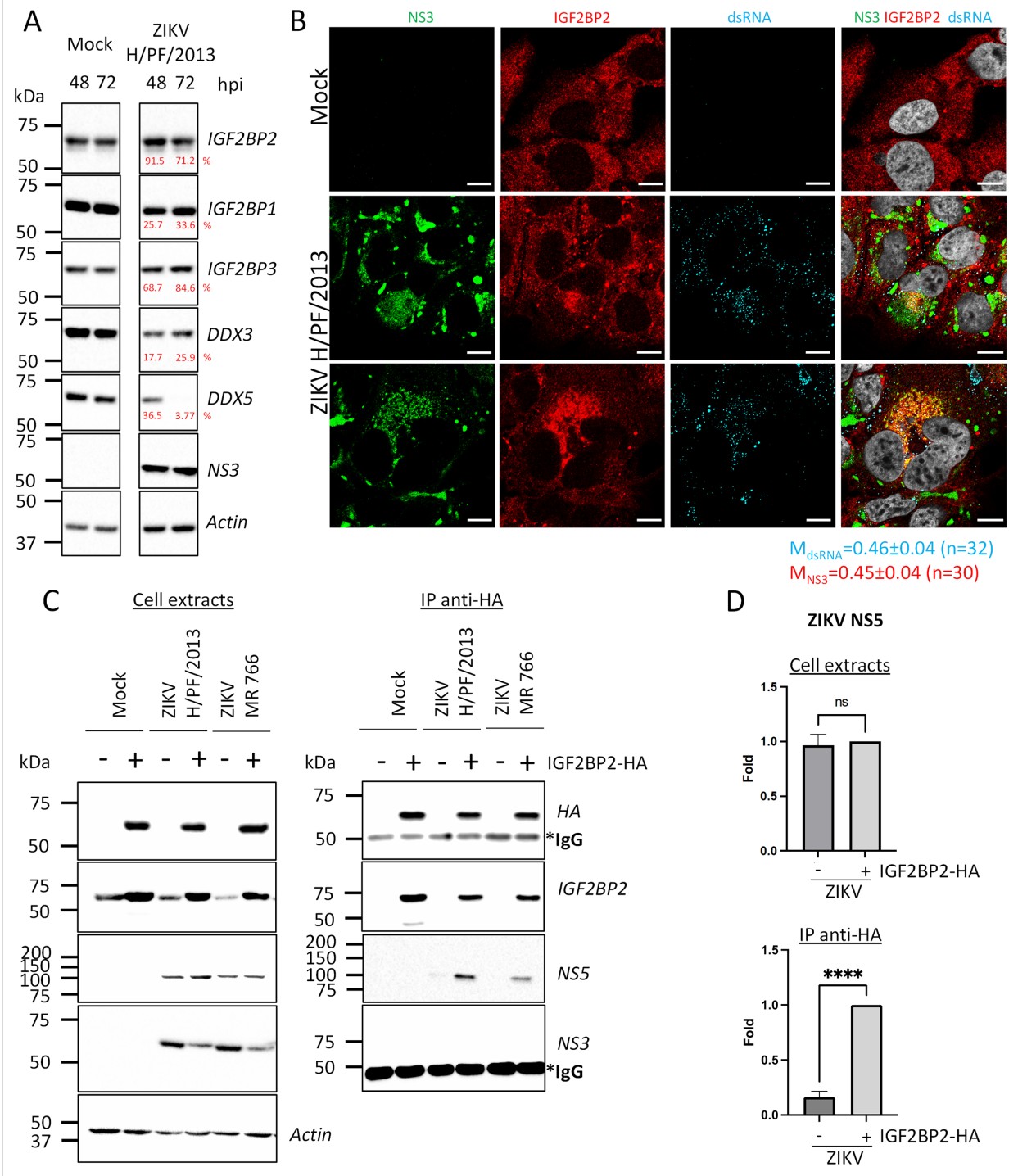

**Figure 4.** IGF2BP2 associates with NS5 and accumulates into Zika virus (ZIKV) replication compartment. (**A**) Huh 7.5 cells were infected with ZIKV H/PF/2013 at an MOI of 5. Cells were collected at 48 and 72 hr post-infection (hpi). Cell extracts were prepared and analyzed by western blotting using the indicated antibodies. Actin-normalized protein signals are shown. (**B**) Huh7.5 cells were infected with ZIKV H/PF/2013 with an MOI of 10 or left uninfected. Two days post-infection, cells were fixed, immunolabeled for the indicated factors, and imaged by confocal microscopy. Scale bar = 10 µm. The Manders' coefficient (mean ± SEM) representing the fraction of dsRNA (cyan) and NS3 (red) signals overlapping with IGF2BP2 signal is shown (n=number of cells). (**C**) Co-immunoprecipitation assays using HA antibodies were performed with extracts from Huh7.5 cells stably expressing IGF2BP2-HA (+) or control-transduced cells (-) which were infected with ZIKV at an MOI of 10 for 2 days. Purified complexes were analyzed for their protein content by western blotting. (**D**) Means of quantified NS5 signals from (C) (normalized to actin [extracts] or IGF2BP2 [IP]) ± SEM are shown based on nine independent experiments. ****: p<0.0001; ns: not significant (unpaired t-test).

*Figure 4 continued on next page*

*Figure 4 continued*

The online version of this article includes the following source data and figure supplement(s) for figure 4:

**Source data 1.** Data points to generate the bar graphs of *Figure 4D* and quantify mean Manders' coefficients of *Figure 4B*.

**Source data 2.** PDF file containing original western blots for *Figure 2A and C*, indicating the relevant bands and conditions.

**Source data 3.** Original files for western blot analysis displayed in *Figure 2A and C*.

**Figure supplement 1.** IGF2BP2 relocalizes to the viral replication compartment in Zika virus (ZIKV)-infected cells at 1 day post-infection.

**Figure supplement 2.** HA-tagged IGF2BP2 relocalizes to the viral replication compartment in Zika virus (ZIKV)-infected cells, as endogenous IGF2BP2.

To assess a potential association between IGF2BP2 and ZIKV vRNA, we first performed protein immunostaining coupled to single RNA molecule fluorescence *in situ* hybridization (FISH) using signal-amplified DNA branched probes to detect IGF2BP2 and ZIKV RNA, respectively. ZIKV RNA was specifically visualized since no FISH signal was detected in uninfected cells while in ZIKV-infected cells, vRNA exhibited a punctate distribution (*Figure 5A*, *Figure 5—figure supplement 1*). Co-staining of vRNA and IGF2BP2 revealed their partial colocalization (*Figure 5A*, white arrows, mean Manders' coefficient = 0.28 ± 0.02 [n=31]), which strongly suggests that IGF2BP2 associates with the ZIKV RNA genome.

To test this, we performed anti-HA co-immunoprecipitation assays using extracts of Huh7.5 cells as in *Figure 4C*. Immune complexes were subjected to RNA purification and subsequently to RT-qPCR to detect viral genome (*Figure 5B*). While the overexpression of IGF2BP2 had no impact on total vRNA levels, we detected a highly significant enrichment of vRNA in purified IGF2BP2-HA complexes compared to the negative specificity control (no IGF2BP2-HA expression). This demonstrates that vRNA and IGF2BP2 are part of the same RNP complex. To further investigate if IGF2BP2 can interact with ZIKV vRNA in a direct manner, we have performed *in vitro* binding assays using microscale thermophoresis (MST) with different regions of ZIKV vRNA and IGF2BP2 protein. We have focused our analysis on vRNA 5' NTR and 3' NTR, which are highly structured and are absolutely required for vRNA synthesis and translation, notably through interactions with viral and host factors (for a review, see *Mazeaud et al., 2018*). Moreover, it has been shown that IGF2BP2 preferentially binds to 3' and 5' untranslated regions of mRNAs (*Zhao et al., 2022*). ZIKV RNA 3' NTR and 5' NTR were synthesized by *in vitro* transcription. In parallel, recombinant IGF2BP2 N-terminal and C-terminal moieties containing either the two RRM (IGF2BP2$_{RRM}$) or both KH3 and KH4 domains (IGF2BP2$_{KH34}$), respectively, were produced in bacteria and subsequently purified (*Figure 5C*). MST revealed that IGF2BP2$_{KH34}$ binds both ZIKV 5' NTR and 3' NTR with high affinity with respective $K_d$ of 203 nM ± 51 and 418 nM ± 49, respectively (*Figure 5D*). In stark contrast, IGF2BP2$_{RRM}$ specifically associated with ZIKV 3' NTR with a lower affinity ($K_d$ = 1598 nM± 257) but this interaction was highly specific since no binding was detected between this recombinant protein and ZIKV 5' NTR. It is noteworthy to mention that we could not assess higher IGF2BP2$_{RRM}$ concentrations than 10 μM to reach binding saturation because of protein aggregation. Overall, these data demonstrate that IGF2BP2 directly and specifically interacts with vRNA in infected cells.

## ZIKV regulates vRNA replication

The fact that IGF2BP2 associates with both ZIKV NS5 RdRp and vRNA led us to hypothesize that IGF2BP2 is involved in the vRNA replication step of ZIKV life cycle. To test this, we took advantage of reporter ZIKV sub-genomic replicons based on the H/PF/2013 genome (ZIKV sgR2A; *Figure 6A*, *Münster et al., 2018*). This engineered genome has been deleted for the coding sequence of structural proteins and expresses Rluc in frame with the NS1-NS5 polyprotein. When *in vitro*-transcribed genomes are introduced in cells by electroporation, they autonomously replicate but neither virus assembly nor entry occur because of the lack of structural proteins. Hence, the luciferase activity in transduced Huh7.5 cells was used as a read-out of vRNA replication 2 days post-electroporation. Strikingly, ZIKV sgR2A replication was attenuated in IGF2BP2 KD, with a significant 40% decrease (*Figure 6B*). To rule out that this phenotype was due to a potential defect in vRNA translation, we used mutated Rluc-expressing ZIKV sub-genomes which express a defective NS5 RdRp and do no replicate (ZIKV sgR2A GAA; *Figure 6A and B*). Hence, the Rluc activity detected at 4 hr post-electroporation entirely relies on vRNA translation. IGF2BP2 KD did not have any significant impact on Rluc activity in Huh7.5 cells transfected with either sgR2A or sgR2A GAA (*Figure 6C*). These data show that IGF2BP2

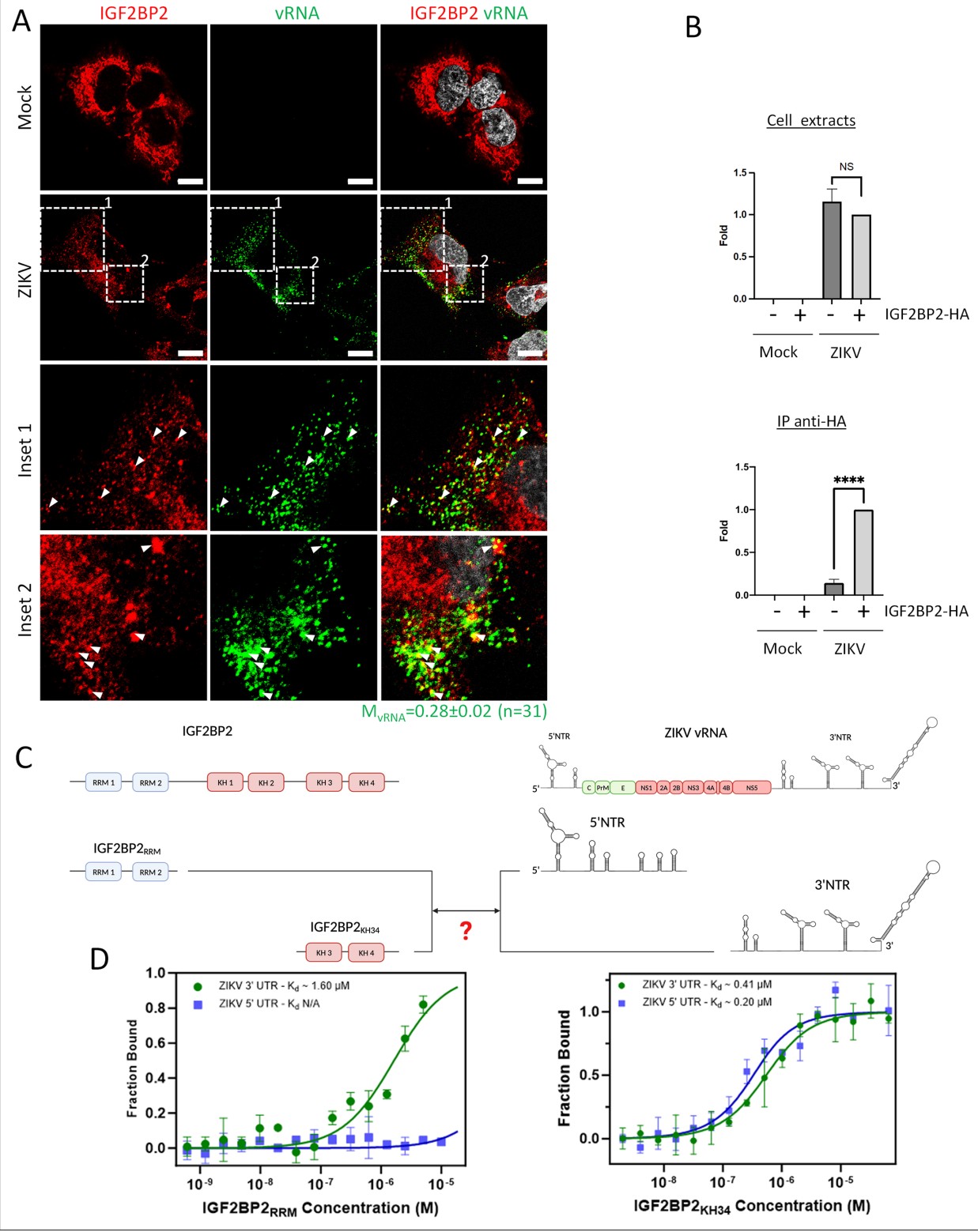

**Figure 5.** IGF2BP2 interacts with Zika virus (ZIKV) viral RNA (vRNA). (**A**) Fluorescence *in situ* hybridization (FISH) and IGF2BP2 immunostaining were performed using Huh 7.5 cells which were infected for 2 days with ZIKV (MOI = 10) or left uninfected. The Manders' coefficient (mean ± SEM) representing the fraction of vRNA signal overlapping with IGF2BP2 signal is shown (n=number of cells). Scale bar = 10 μm. (**B**) Huh7.5 cells expressing IGF2BP2-HA and control cells were infected with ZIKV H/PF/2013 at an MOI of 10, or left uninfected. Two days later, cell extracts were prepared and subjected to anti-HA immunoprecipitations. Extracted vRNA levels were measured by RT-qPCR. Means ± SEM are shown based on three independent

*Figure 5 continued on next page*

*Figure 5 continued*

experiments. ****: p<0.0001; NS: not significant (unpaired t-test). (**C**) IGF2BP2 recombinant proteins containing either the two RNA recognition motifs (RRM) or KH3 and KH4 domains were produced in bacteria and purified. In parallel ZIKV 5' nontranslated region (NTR) and 3' NTR were synthesized by *in vitro* transcription. (**D**) Combination of truncated IGF2BP2 proteins and either ZIKV 5' NTR (blue squares) or ZIKV 3' NTR (green circles) were used for *in vitro* binding assays using microscale thermophoresis.

The online version of this article includes the following source data and figure supplement(s) for figure 5:

**Source data 1.** Data points to generate the bar graphs and binding curves of *Figure 5B and D*, and quantify the mean Manders' coefficient of *Figure 5A*.

**Figure supplement 1.** IGF2BP2 partially colocalizes with NS3 and viral RNA (vRNA) in Zika virus (ZIKV)-infected cells.

positively regulates ZIKV genome replication (but not its translation), which is most likely mediated through its interactions with NS5 and vRNA.

## ZIKV infection modulates the interactions between IGF2BP2 and its endogenous mRNA ligands

Since IGF2BP2 associates with NS5 and vRNA and accumulates in the viral replication compartment, we hypothesized that ZIKV infection induces a remodeling of IGF2BP2 RNP, notably regarding its content in endogenous mRNA partners. To test this, we performed immunopurifications of IGF2BP2 complexes followed by RT-qPCR as in *Figures 4C and 5B* in order to determine the relative abundance of three known IGF2BP2 mRNA ligands (namely *TNRC6A*, *PUM2*, and *CIRBP*) (*Huang et al., 2018*) in both control and ZIKV-infected cells. *TRNC6A*, *PUM2*, and *CIRBP* mRNAs were selected because ZIKV is known to alter their $N^6$-adenosine methylation status (*Gokhale et al., 2020*), an epitranscriptomic RNA modification that increases the affinity for IGF2BP2, a known 'm6A reader' (*Huang et al., 2018*). As expected, we could specifically detect these mRNA in purified IGF2BP2 complexes in uninfected cells (Mock+IGF2BP2-HA condition; *Figure 7A–C*). Interestingly, the levels of co-immunoprecipitated *TNRC6A* and *PUM2* mRNAs cell were significantly decreased when Huh7.5 were infected with ZIKV while total mRNA levels remained unchanged (*Figure 7A and B*), suggesting that ZIKV induced a loss of binding between IGF2BP2 and these endogenous mRNAs. In contrast, no significant changes in IGF2BP2 association were observed for *CIRBP* mRNA in either condition although it is noteworthy that total levels were increased in total extracts when IGF2BP2 was overexpressed (*Figure 7C*). These results show that ZIKV infection modifies the interaction between IGF2BP2 and specific endogenous mRNAs and is consistent with the notion of a virus-induced remodeling of IGF2BP2 RNP.

## ZIKV alters IGF2BP2 proteo-interactome in infected cells

Next, to assess our hypothesis of a ZIKV-remodeled IGF2BP2 RNP, we investigated whether ZIKV infection changes the protein interaction profile of IGF2BP2. We first performed western blotting on purified IGF2BP2-HA samples (prepared as in *Figures 4C and 5B*) to detect IGF2BP1, IGF2BP3, and YBX1 which are known IGF2BP2 partners (*Chatel-Chaix et al., 2013*; *Jønson et al., 2007*; *Nielsen et al., 2004*). Comparable amounts of either partner were specifically co-purified with IGF2BP2-HA (*Figure 8A–D*), indicating that ZIKV infection does not significantly change their association. As expected from results shown in *Figure 4C*, the viral RdRp NS5 was readily detected in this complex. Interestingly, confocal microscopy showed that ZIKV infection induced an accumulation of IGF2BP1, IGF2BP3, and YBX1 in the replication compartment as for IGF2BP2 with partial colocalization with dsRNA and NS3 (*Figure 8—figure supplements 1 and 2C, D*), which supports that ZIKV physically co-opts an RNP containing these fours RBPs. This phenotype was specific to these host factors as it was not observed for the RBPs DDX5 and LARP1 (*Figure 8—figure supplement 2A–D*). Consistent with the fact that these proteins (including NS5) belong to a ribonucleoprotein complex containing IGF2BP2, their interaction with IGF2BP2 was RNA-dependent since they were barely detectable in the immunoprecipitates when the cell extracts were treated with RNase A prior to anti-HA pull-down (*Figure 8—figure supplement 3*). To globally evaluate changes in the IGF2BP2 interactome upon infection, we analyzed the protein composition of IGF2BP2-HA complexes immunopurified from uninfected cells or infected with ZIKV or DENV by mass spectrometry (MS). We identified 527 proteins which specifically interacted with IGF2BP2 (*Figure 9—figure supplement 1A and B*; *Supplementary file 1*). The abundance of over 86% of the proteins in IGF2BP2-HA complexes (455 proteins),

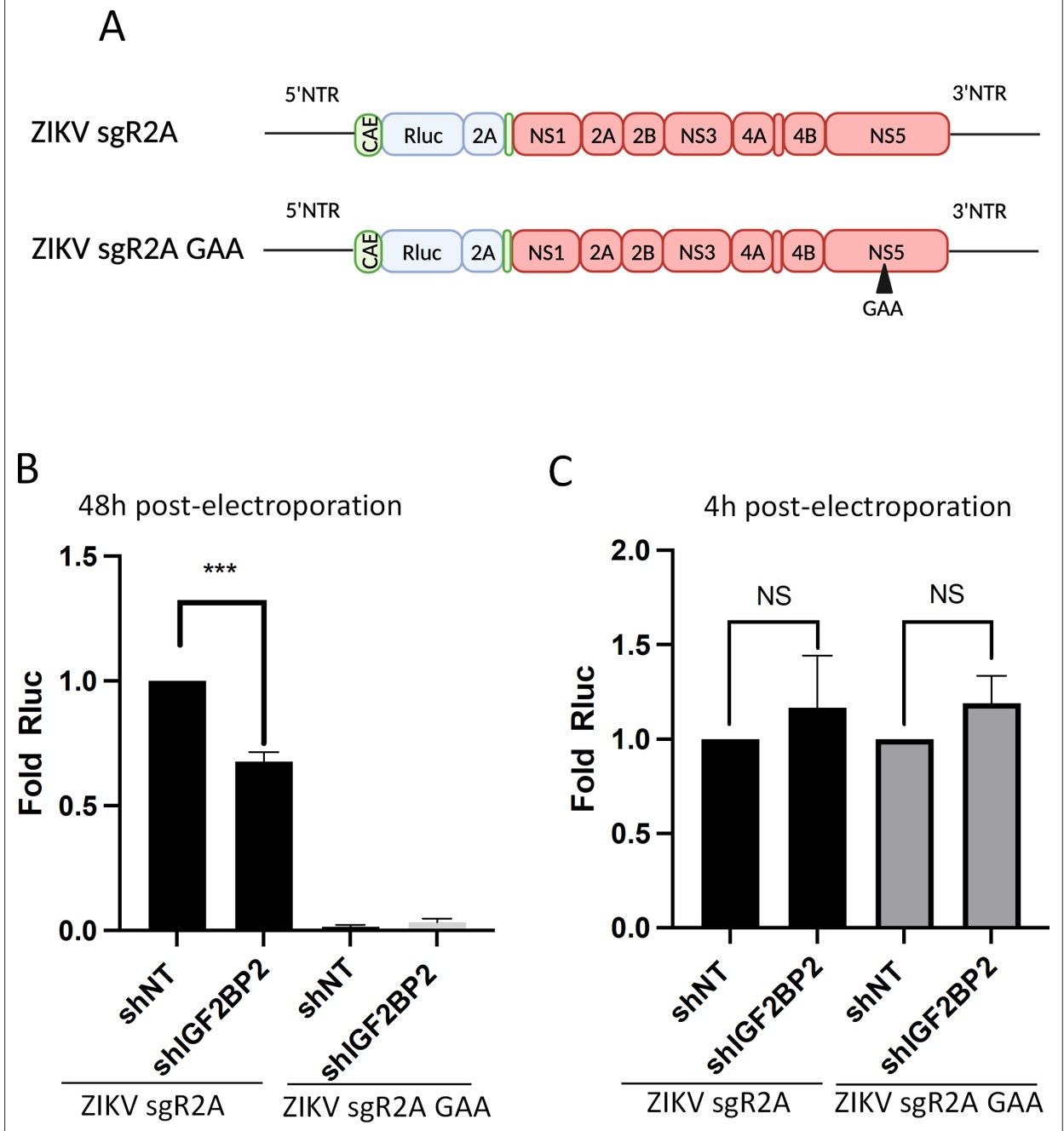

**Figure 6.** IGF2BP2 regulates the replication of Zika virus (ZIKV) viral RNA (vRNA). (**A**) Schematic representation of reporter ZIKV H/PF/2013 sub-genomic replicons (sgR2A) and replication-deficient genomes because of mutations in NS5 RNA-dependent RNA polymerase (RdRp) sequence (sgR2A GAA). (**B–C**) Huh7.5 were transduced with short-hairpin RNA (shRNA)-expressing lentiviruses and subjected to electroporation with *in vitro*-transcribed sgR2A or sgR2A GAA RNAs 2 days later. In-cell bioluminescence was measured (**B**) 48 or (**C**) 4 hr post-electroporation and normalized to the non-target shRNA (shNT) control condition. In (C), the luciferase activity was normalized to the transfection efficiency, i.e., the Renilla luciferase (Rluc) activity at 4 hr post-electroporation. Means ± SEM are shown based on four independent experiments. ***: p<0.001; NS: not significant (unpaired t-test).

The online version of this article includes the following source data for figure 6:

**Source data 1.** Data points to generate all bar graphs of *Figure 6*.

including IGF2BP1, IGF2BP3, and YBX1, remained unchanged upon either infection (as expected from *Figure 8A–D*). In contrast, in ZIKV-infected cells, the interactions of IGF2BP2 with 40 and 22 proteins were either decreased or increased, respectively, with 52 of these changes being specifically observed only in ZIKV-infected cells (*Figure 9A*; *Figure 9—figure supplement 1A*). Gene ontology

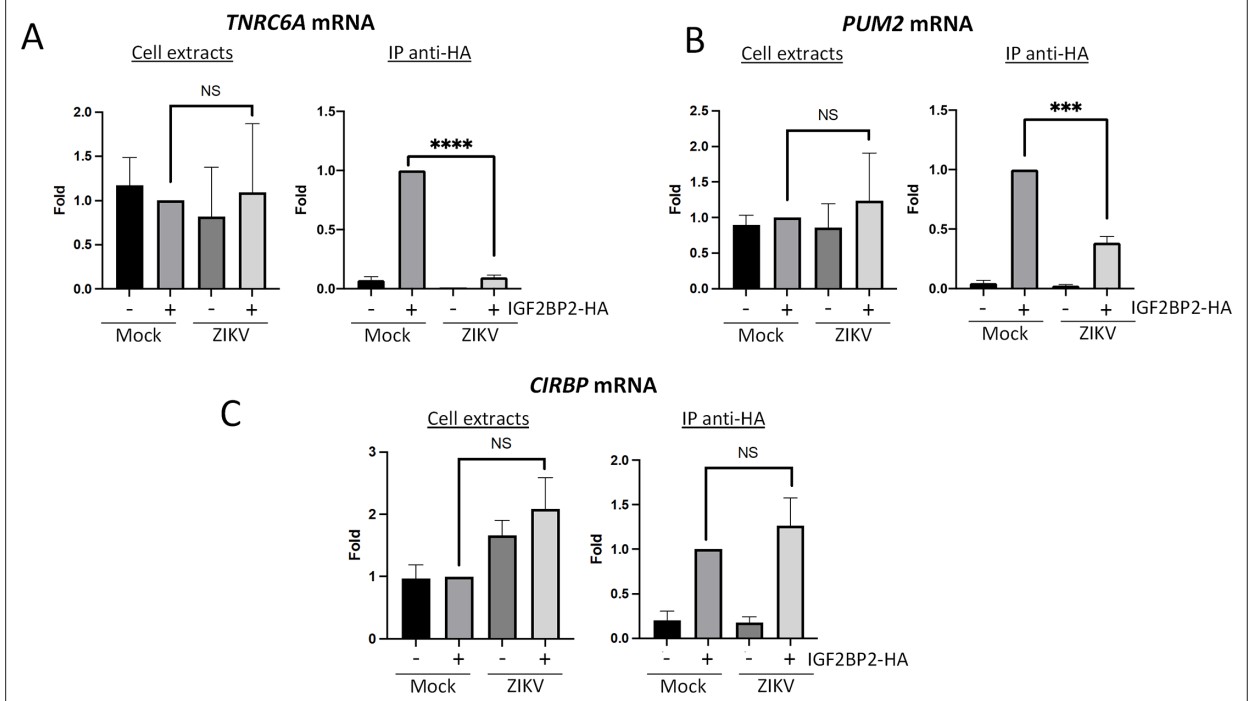

**Figure 7.** Zika virus (ZIKV) infection decreases the interaction between IGF2BP2 and several of its mRNA endogenous ligands. Huh7.5 cells stably expressing IGF2BP2-HA (+) and control cells (-) were infected with ZIKV H/PF/2013 at an MOI of 10, or left uninfected. Two days later, cell extracts were prepared and subjected to anti-HA immunoprecipitations. Extracted (**A**) *TNRC6A*, (**B**) *PUM2*, and (**C**) *CIRBP* mRNA levels were measured by RT-qPCR. Means ± SEM are shown based on three independent experiments. ****: p<0.0001; ***: p<0.001; NS: not significant (unpaired t-test).

The online version of this article includes the following source data for figure 7:

**Source data 1.** Data points to generate all bar graphs of *Figure 7*.

analysis of the 62 partners whose interaction with IGF2BP2 was altered during ZIKV infection revealed a high enrichment of biological processes related to mRNA splicing (*Figure 9B*). The interactome tree of these 62 IGF2BP2 protein partners generated with STRING database highlighted two major clusters regulated by ZIKV (*Figure 9C*). In line with our gene ontology analysis, one of these clusters comprised proteins involved in mRNA splicing for most of which interaction with IGF2BP2 was significantly decreased during ZIKV infection (red circles). The second identified cluster was related to ribosome biogenesis.

One of the best partners specifically modulated by ZIKV and/or DENV was Atlastin 2 (ATL2) whose interaction with IGF2BP2 was most significantly increased upon ZIKV infection (p-value=$10^{-5.7}$; ***Supplementary file 1***; highlighted with * in *Figure 9C* and *Figure 9—figure supplement 1B*). ATL2 is an ER-shaping protein (***Wang et al., 2016***) which was reported to be involved in the formation of orthoflavivirus VPs (***Neufeldt et al., 2019***). We validated this ZIKV-specific phenotype by co-immunoprecipitation assays (*Figure 9—figure supplement 2A and B*, *Figure 8—figure supplement 3A*) in which we could detect a 14-fold increase in IGF2BP2-HA/ATL2 interaction when cells were infected with ZIKV compared to uninfected or DENV-infected samples. As an additional specificity control, we did not detect any interaction between ATL2 and HA-tagged VCP (*Figure 9—figure supplement 2*), an ER quality control protein which was shown to be physically recruited into DENV and ZIKV replication compartments (***Anton et al., 2021***; ***Mazeaud et al., 2021***), ruling out that ATL2/IGF2BP2 interaction is simply due to the presence of ATL2 within remodeled ER or because of non-specific binding to the HA-tag. Altogether, these data show that ZIKV infection specifically alter the composition of IGF2BP2 RNP complex.

## IGF2BP2 is involved in the biogenesis of ZIKV replication organelles

The fact that ZIKV infection promotes IGF2BP2 association with ATL2 led us to hypothesize that IGF2BP2 regulates vRNA replication by contributing to ZIKV replication organelle biogenesis. Since

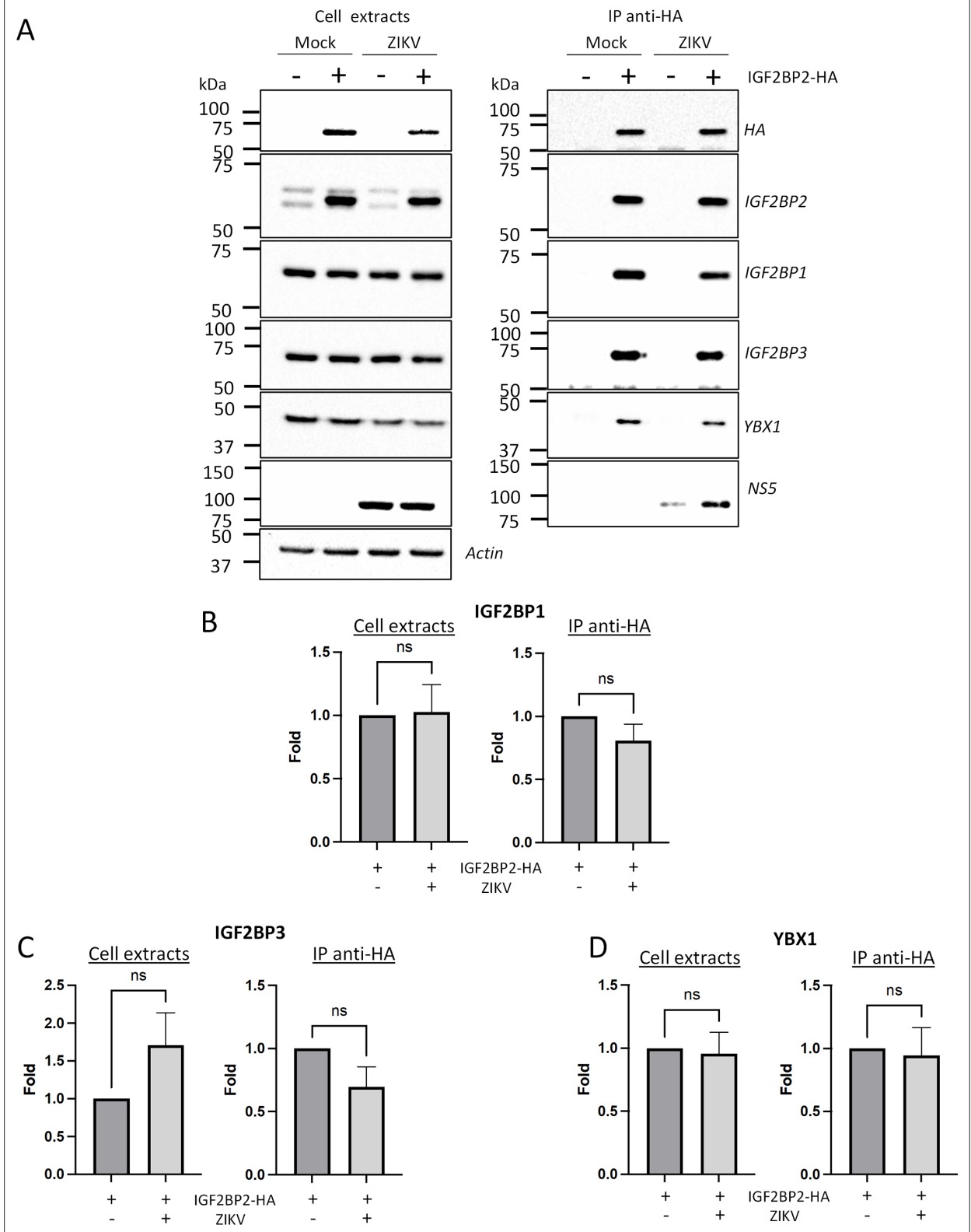

**Figure 8.** Zika virus (ZIKV) infection does not significantly impact the association of IGF2BP2 with IGF2BP1, IGF2BP3, and YBX1. Huh7.5 cells stably expressing IGF2BP2-HA (+) and control cells (-) were infected with ZIKV H/PF/2013 at an MOI of 10, or left uninfected. Two days later, cell extracts were prepared and subjected to anti-HA immunoprecipitations. (**A**) Purified complexes were analyzed by western blotting for their content in the indicated

*Figure 8 continued on next page*

*Figure 8 continued*

proteins. IGF2BP1 (**B**), IGF2BP3 (**C**), and YBX1 (**D**) levels were quantified and means of protein signals (normalized to actin [extracts] and IGF2BP2 [IP]) ± SEM are shown based on six to eight independent experiments. ns: not significant (unpaired t-test).

The online version of this article includes the following source data and figure supplement(s) for figure 8:

**Source data 1.** Data points to generate the bar graphs of *Figure 7B–D*.

**Source data 2.** PDF file containing original western blots for *Figure 8A*, indicating the relevant bands and conditions.

**Source data 3.** Original files for western blot analysis displayed in *Figure 8A*.

**Figure supplement 1.** IGF2BP1, IGF2BP3, and YBX1 relocalize to the viral replication compartment in Zika virus (ZIKV)-infected cells.

**Figure supplement 1—source data 1.** Data points to determine the mean Manders' coefficients.

**Figure supplement 2.** LARP1 and DDX5 do not relocalize to the viral replication compartment in Zika virus (ZIKV)-infected cells.

**Figure supplement 2—source data 1.** Data points to determine the mean Manders' coefficients.

**Figure supplement 3.** The association between IGF2BP2 and Zika virus (ZIKV) NS5 is RNA-dependent in infected cells.

**Figure supplement 3—source data 1.** Data points to generate the bar graphs in *Figure 8—figure supplement 3C* .

**Figure supplement 3—source data 2.** PDF file containing original western blots for panel A, indicating the relevant bands and conditions.

**Figure supplement 3—source data 3.** Original files for western blot analysis displayed in panel A.

IGF2BP2 KD decreases vRNA replication and thus, indirectly reduces the overall synthesis of viral proteins that drive vRO formation, evaluating this hypothesis in infected cells was not possible. To tackle this challenge, we took advantage of a recently described plasmid-based system (named pIRO-Z) (*Cerikan et al., 2020*; *Goellner et al., 2020*) which induces vROs *de novo* in transfected cells in a replication-independent manner (*Figure 10A*). This plasmid allows the cytoplasmic transcription of NS1-NS5 polyprotein-encoding mRNA under the control of T7 RNA polymerase which is stably overexpressed in Huh7-derived Lunet-T7 cells.

Lunet-T7 cells were transduced with shNT/shIGF2BP2-expressing lentiviruses and subsequently transfected with pIRO-Z. The efficiency of IGF2BP2 KD was controlled by RT-qPCR (*Figure 10B*). Confocal microscopy of NS3-immunolabeled cells confirmed that the transfection efficiencies were comparable between all conditions (*Figure 10C and D*). When imaged by electron microscopy (EM), VPs were detected in 70% of the shNT-transduced cells. In contrast, IGF2BP2 KD resulted in a decrease in the proportion of cells containing vROs (*Figure 10E and F*) while the diameter of VPs remained unchanged (*Figure 10G*). These results demonstrate that IGF2BP2 plays a role in the biogenesis of ZIKV VPs independently from vRNA synthesis.

## Discussion

In this study, to identify new host factors that regulate viral processes involving the orthoflavivirus vRNA, we have assessed 10 RBPs which were previously reported to regulate the life cycle of HCV, an hepacivirus belonging to the same *Flaviviridae* family as DENV and ZIKV (*Chatel-Chaix et al., 2013*; *Chatel-Chaix et al., 2011*). For instance, YBX1, IGF2BP2, LARP1, DDX6, C1QBP1 were reported to positively regulate HCV RNA synthesis while inhibiting the production of infectious viral particles. Our RNAi mini-screen assessing the impact of gene depletion on DENV or ZIKV infection revealed that overall, such host factor-mediated regulations are not fully conserved across the *Flaviviridae* family. Only LARP1 KD reduced ZIKV replication while it increased virus production as observed for HCV (*Chatel-Chaix et al., 2013*). A comparable trend was observed for YBX1 although the differences were not significant. In case of DENV, none of the tested RBPs increased infectious particle production. The fact that the phenotypes did not fully mirror those we have reported for HCV might reflect the differences in the architecture of the vROs induced by orthoflaviviruses (i.e. invaginated vesicles in the ER; *Cortese et al., 2017*; *Gillespie et al., 2010*; *Miorin et al., 2013*; *Welsch et al., 2009*; *Westaway et al., 1997*) compared to that of HCV double-membrane vesicles which result from ER protrusions (*Romero-Brey et al., 2012*). The best observed phenotype in our RNAi mini-screen was observed with IGF2BP2 KD which inhibited the replication of ZIKV but not that of other tested orthoflaviviruses or SARS-CoV-2, another positive-strand RNA virus from the *Coronaviridae* family. We further observed that ZIKV induced a relocalization of IGF2BP2 to the replication compartment in which it associates with NS5 and vRNA. Consistently, IGF2BP2 KD decreased the efficiency of vRNA

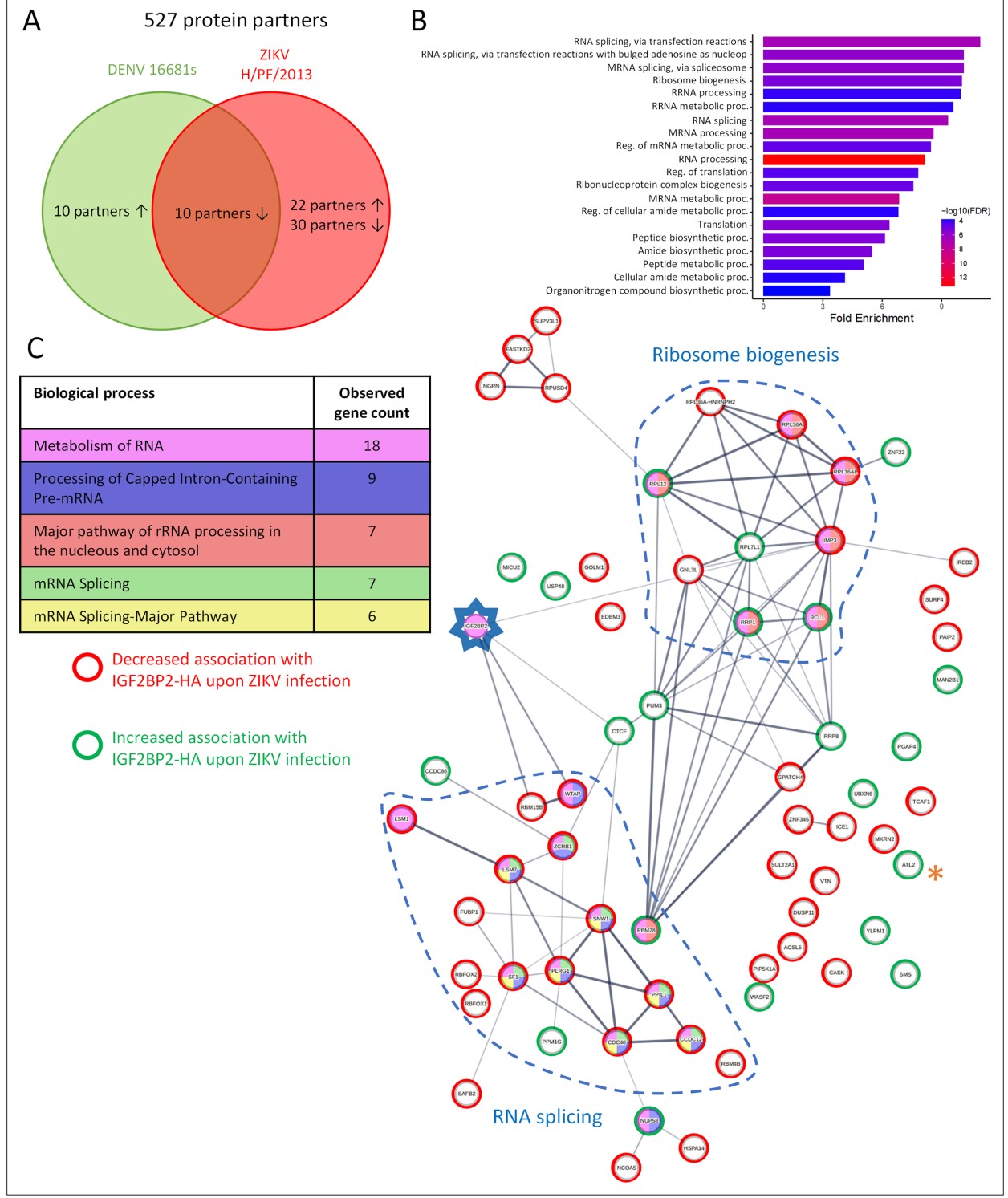

**Figure 9.** Zika virus (ZIKV) infection alters the IGF2BP2 proteo-interactome. Huh7.5 cells expressing IGF2BP2-HA and control cells were infected with ZIKV H/PF/2013, dengue virus serotype 2 (DENV2) 16681s, or left uninfected. Two days later, cell extracts were prepared and subjected to anti-HA immunoprecipitations. Resulting complexes were analyzed by quantitative mass spectrometry. (**A**) Venn diagram depicting the overlap between IGF2BP2 partners modulated by ZIKV and/or DENV infections. (**B**) Gene ontology (GO) biological process analyses of the IGF2BP2 interactions which were impacted upon ZIKV infection. (**C**) Interaction tree of the 62 IGF2BP2 interactions modulated by ZIKV infection (generated with STRING online resource). The red and green circles identify the partners of the STRING network whose association with IGF2BP2 is decreased and increased during infection, respectively. The biological process analysis generated by STRING is also shown.

*Figure 9 continued on next page*

*Figure 9 continued*

The online version of this article includes the following source data and figure supplement(s) for figure 9:

**Figure supplement 1.** Orthoflavivirus infection modulates the interactome of IGF2BP2.

**Figure supplement 2.** Zika virus (ZIKV) infection enhances the interaction between IGF2BP2 and ATL2.

**Figure supplement 2—source data 1.** Data points to generate the bar graphs in *Figure 9—figure supplement 2B*.

**Figure supplement 2—source data 2.** PDF file containing original western blots for panel A, indicating the relevant bands and conditions.

**Figure supplement 2—source data 3.** Original files for western blot analysis displayed in panel A.

synthesis (but not its translation) correlating with an impairment in VP formation (*Figures 6 and 10*). This highlights that IGF2BP2 is an important regulator of the vRNA replication step of ZIKV life cycle. Of note, the overexpression did not result in an increase of vRNA abundance. We believe that this is because endogenous IGF2BP2 is highly expressed in cancer cells such as the Huh7.5 cells used here and is presumably not limiting for viral replication in this system (*Cui et al., 2021*; *Feng et al., 2022*; *Liu et al., 2023a*; *Pu et al., 2020*; *Zhang et al., 2024*).

Early in the ZIKV replication cycle, newly produced viral nonstructural proteins induce ER alterations to form VPs which will host the vRNA synthesis machinery including NS5 RNA polymerase (*Neufeldt et al., 2018*). However, the molecular mechanisms governing VP morphogenesis remain poorly understood, mostly because it is experimentally challenging to discriminate phenotypes related to vRO morphogenesis from those resulting from perturbations in vRNA replication which reduce viral protein input and hence, VP abundance indirectly. In this study, using a recently engineered replication-independent VP induction system (*Cerikan et al., 2020*; *Goellner et al., 2020*), we identified IGF2BP2 as a novel regulator of ZIKV *de novo* VP biogenesis. Interestingly, IGF2BP2 associates with ZIKV RNA 3' NTR and ATL2, which are both co-factors of VP formation (*Cerikan et al., 2020*; *Neufeldt et al., 2019*), suggesting that they all regulate this process as part of the same RNP complex. *In vitro* RNA-binding assays showed that IGF2BP2 can interact directly and specifically with vRNA 3' NTR via its RRM domains. A recombinant protein comprising the third and fourth KH domains of IGF2BP2 also bound 3' NTR. However, this interaction appeared less specific since a comparable affinity was observed for the 5' NTR *in vitro*. Besides, we cannot exclude that IGF2BP2 binds additional vRNA regions in the coding sequence, which was not tested in this study. Interestingly, it has been demonstrated that the KH domains of IGF2BPs mediate the binding to $N^6$-methylated adenosines in mRNA (*Huang et al., 2018*), a modification that has been detected within the RNA genome of ZIKV and other Flavividae viruses, including in the 3' NTR (*Gokhale et al., 2016*; *Lichinchi et al., 2016*; *McIntyre et al., 2018*; *Ruggieri et al., 2021*). Since IGF2BP2 is a 'm6A reader' through KH3 and KH4 (*Huang et al., 2018*; *Li et al., 2019*), it is highly plausible that this RNA modification contributes to the vRNA-binding specificity and the proviral roles of this host factor. Considering that the binding of KH3 and KH4 of IGF2BP1 paralogue to RNA can induce conformational changes of the RNA (*Patel et al., 2012*), it will be interesting to evaluate whether IGF2BP2 co-opting by replication complexes (most likely via NS5) contributes to the structural switches and/or secondary and tertiary structures of vRNA controlling its fate during the life cycle.

To study the impact of ZIKV infection on IGF2BP2 association with endogenous RNAs, we decided to focus our analysis on three known IGF2BP2 mRNA ligands namely *CIRBP*, *TNRC6A*, and *PUM2*. We chose to focus on those mRNA because: (i) ZIKV, in addition to other *Flaviviridae* viruses (DENV, WNV, and HCV), alter their $N^6$-adenosine methylation status during infection (*Gokhale et al., 2016*; *Lichinchi et al., 2016*; *McIntyre et al., 2018*), and (ii) IGF2BP2 is a well-described 'm6A reader' which regulates the fate of bound RNAs (*Huang et al., 2018*; *Li et al., 2019*). More specifically, it has been shown that ZIKV infection decreases m6A content of *CIRBP* mRNA and increases the one of *PUM2* and *TNRC6A* mRNAs. Unexpectedly, we observed a decreased interaction between these overmethylated mRNAs and IGF2BP2 while IGF2BP2/*CIRBP* mRNA association remained unchanged (*Figure 7*). This counterintuitive observation might be due to additional layers of IGF2BP2 regulation such as potential virus-dependent post-translational modifications. Indeed, the mTOR complex 1 (mTORC1) was reported to phosphorylate two residues of IGF2BP2 (Ser162/Ser164) which are located between the second RRM domain and the first KH domain (*Dai et al., 2011*). The simultaneous phosphorylation of these two serine residues strongly enhances the interaction between IGF2BP2 and IGF-II leader 3 mRNA 5'UTR. Interestingly, our preliminary MS analysis of IGF2BP2 phosphopeptides abundance indicates that ZIKV

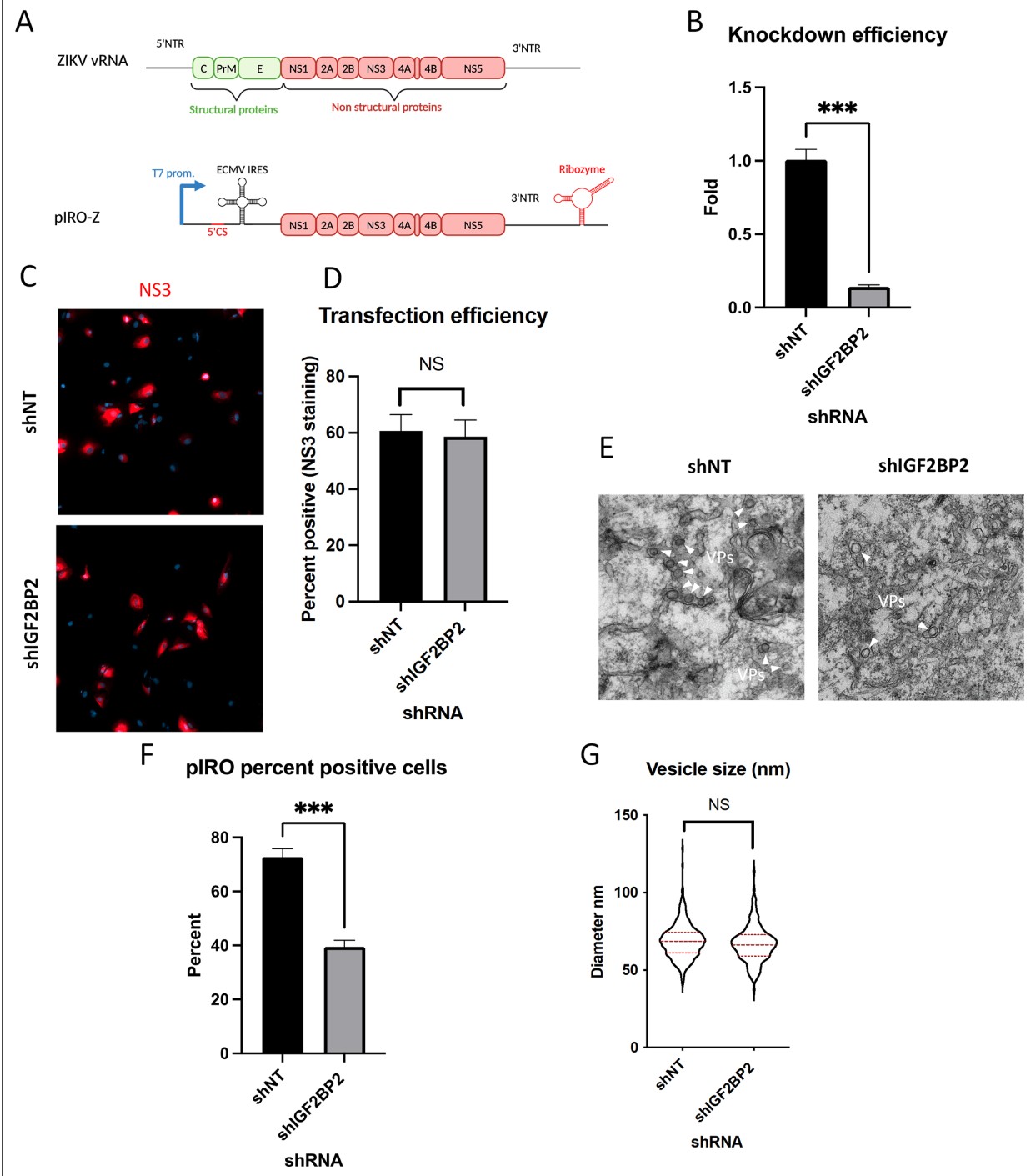

**Figure 10.** IGF2BP2 regulates the biogenesis of Zika virus (ZIKV) replication organelles. (**A**) Schematic representation of the pIRO system. Upon transfection in cells expressing the T7 RNA polymerase, this plasmid allows the cytoplasmic transcription of NS1-NS5 polyprotein under the control of T7 promoter, in a ZIKV replication-independent manner. NS1-5 polyprotein synthesis is under the control of ECMV IRES. The presence of both ZIKV 3' NTR and 5' cyclization sequence (5' CS) is required for efficient vesicle packet (VP) induction. Finally, the activity of HDV ribozyme ensures that the 3' terminus of the RNA is similar to that of viral RNA (vRNA) genome. Huh7-Lunet-T7 were transduced with short-hairpin RNA (shRNA)-expressing lentiviruses at an MOI of 5–10. Two days later, transduced cells were transfected with pIRO-Z plasmid. Sixteen hours later, cells were analyzed for (**B**) IGF2BP2 mRNA levels by RT-qPCR to measure knockdown efficiency, (**C–D**) transfection efficiency by confocal imaging of NS3-labeled cells, and (**E**) for VP content by transmission electron microscopy. Electron micrographs were used to measure (**F**) the percentage of cells with VPs and (**G**) the diameter of VPs in each condition. ***: p<0.001; NS: not significant (unpaired t-test).

The online version of this article includes the following source data for figure 10:

**Source data 1.** Data points to generate all graphs of *Figure 10*.

decreases phosphorylation of Ser162, but not that of Ser164. Moreover, several studies reported that ZIKV infection inhibits mTORC1 signaling (*Liang et al., 2016*; *Liu et al., 2023b*). Thus, it is tempting to speculate that ZIKV, through the mTOR pathway, regulates the affinity of IGF2BP2 for endogenous mRNAs by modulating its phosphorylation status.

Our interactome analysis revealed global changes in the composition of IGF2BP2 RNP with 62 cellular proteins whose association with IGF2BP2 was altered during ZIKV infection. Most notably, ZIKV infection decreased IGF2BP2 interaction with 15 proteins of the mRNA splicing machinery. Interestingly, there is evidence that IGF2BP2 regulates mRNA splicing (*Zhao et al., 2022*) in addition to its most characterized functions in mRNA stability and translation. Furthermore, several studies reported that ZIKV infection alters mRNA splicing (*Bonenfant et al., 2020*; *Hu et al., 2017*; *Michalski et al., 2019*). It was proposed that the expression of the sub-genomic flavivirus RNA (sfRNA), a viral non-coding RNA comprising vRNA 3′ NTR (and hence, predicted to associate with IGF2BP2 as ZIKV genome), alters splicing efficiency by sequestering splicing factors SF3B1 and PHAX (*Michalski et al., 2019*). In case of DENV, NS5 was shown to associate with core components of the U5 snRNP particle and to also modulate splicing (*Maio et al., 2016*). If such activity is conserved for ZIKV NS5, it is tempting to speculate that mRNA splicing is regulated in ZIKV-infected cells by an IGF2BP2/NS5/sfRNA complex. We have also demonstrated that ZIKV infection specifically induces the association between IGF2BP2 and ATL2, an ER-shaping protein which was previously reported to be required for DENV and ZIKV replication (*Monel et al., 2019*; *Neufeldt et al., 2019*). The fact that the KD of IGF2BP2 and ATL2 (this study and *Neufeldt et al., 2019*) both impaired vRO biogenesis suggests that this process involves this ZIKV-specific ATL2-IGF2BP2 complex. Considering that 3′ NTR is an important co-factor of VP formation (*Cerikan et al., 2020*), this complex might contribute to enriching vRNA to VPs through IGF2BP2/3′ NTR interaction. It is noteworthy to mention that in DENV-infected cells, IGF2BP2 is relocalized to the replication compartment and associates with vRNA (unpublished data) but not with ATL2 (*Figure 9—figure supplement 2A–B*). However, IGF2BP2 KD had very little or no effect on all tested DENV strains in contrast to ZIKV and despite our multiple attempts, we could not demonstrate evidence of a specific interaction between IGF2BP2 and DENV NS5. This suggests that (i) ATL2 and NS5 are not directly involved in IGF2BP2 physical hijacking and RNA-binding activity, (ii) the physical co-opting of IGF2BP2 is not sufficient to confer the proviral activity of this host factor. However, ATL2 was reported to regulate DENV replication (*Neufeldt et al., 2019*) even if it does not associate with IGF2BP2 in that infection context. Interestingly, ATL2 interactome analysis in uninfected and DENV-infected cells showed that it associates RBMX and HNRNPC, two other known 'm6A readers' (*Bell et al., 2013*; *Liu et al., 2015*). Intriguingly, ATL2/RBMX interaction was detected only in DENV-infected cells, suggesting that it is induced by the infection and might play a similar role as the one of IGF2BP2/ATL2 in case of DENV. Finally, the interaction of IGF2BP2 with both NS5 and ATL2 was RNA-dependent (*Figure 8—figure supplement 3*), which is not surprising as RNA is often a structural component of ribonucleoprotein complexes. This further supports the idea that IGF2BP2 associates with NS5 and the ER after its binding to the vRNA.

The results described in this study allows the elaboration of a model for IGF2BP2 involvement in ZIKV life cycle (*Figure 11*). In this model, very early in the life cycle, IGF2BP2 associates with vRNA and subsequently to newly synthesized NS5 (step 1). This correlates with changes in the protein composition of IGF2BP2 RNP, including induced interactions with ATL2, which target NS5/vRNA complex to the ER (step 2). Via these changes in the stoichiometry of IGF2BP2 RNP, specific endogenous IGF2BP2 mRNA ligands such as *PUM2* and *TNRC6A* mRNAs are excluded from IGF2BP2 RNP. Subsequently, this ER-bound complex induces morphological alterations of the ER membrane with the contribution of vRNA 3′ NTR and the ER-shaping activity of ATL2 to generate VPs (step 3). In addition to providing an optimal environment for vRNA synthesis, the VPs are believed to contribute to the spatial segregation of the replication complexes, assembling particles and the translation machinery, hence regulating the equilibrium between the multiple fates of vRNA. With that in mind, we cannot exclude that IGF2BP2 contributes to the coordination between replication and assembly by targeting vRNA to/through the VP pore for selective encapsidation into particles budding into the ER (step 4).

Overall, this study highlights the physical and functional interplay between ZIKV replication machinery and IGF2BP2 RNP in infected cells. Considering the diverse roles of IGF2BP2 in mRNA metabolism, it is tantalizing to speculate that the physical and functional hijacking of IGF2BP2 RNP contributes to ZIKV pathogenesis, including developmental defects of the fetal brain.

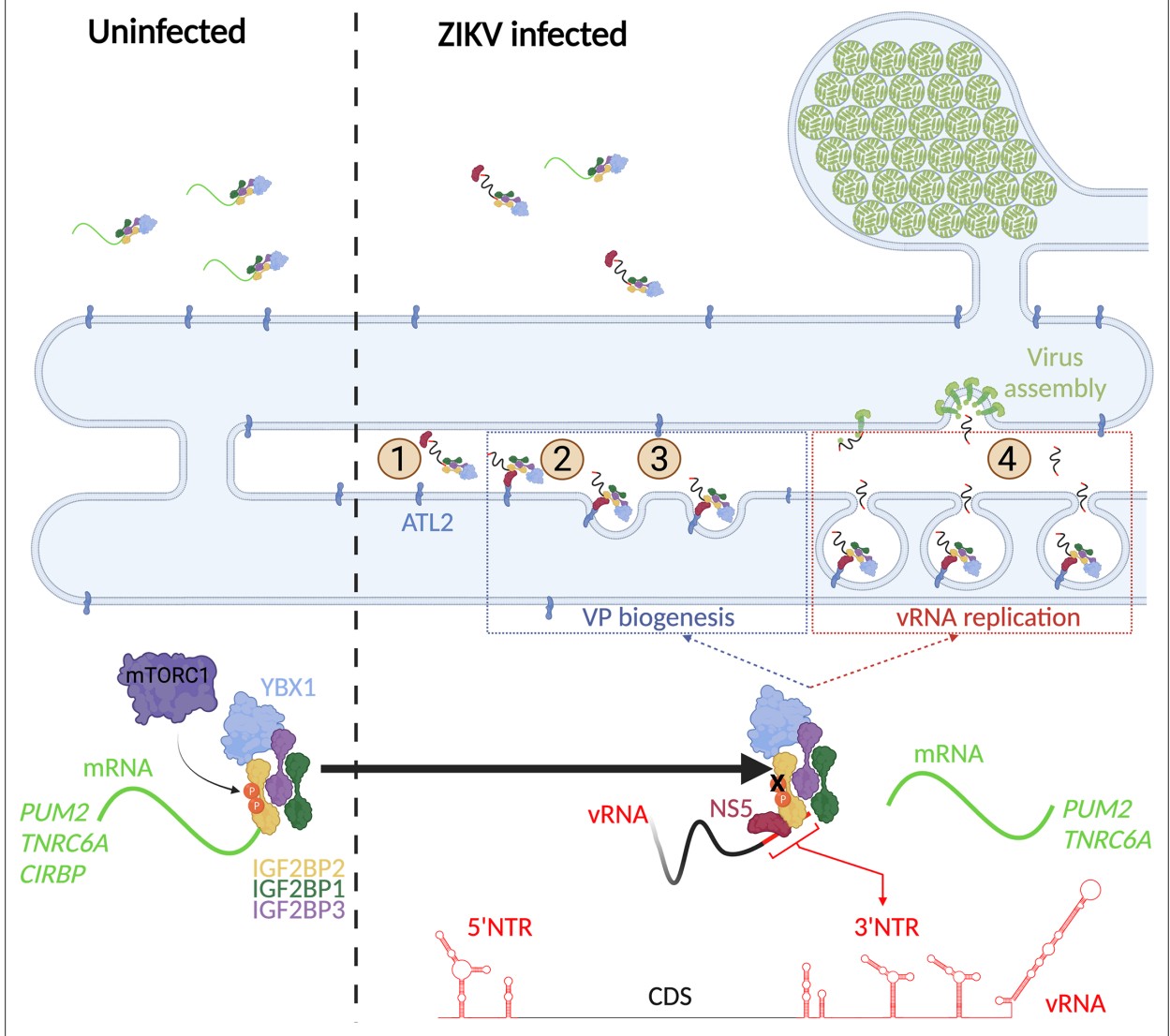

**Figure 11.** A model for IGF2BP2 involvement in Zika virus (ZIKV) life cycle. Step 1: After NS protein synthesis early after virus entry, IGF2BP2 associates with NS5 and vRNA, thus excluding *PUM2* and *TNRC6A* mRNA from the ribonucleprotein (RNP). Step 2: The infection-induced association between IGF2BP2 RNP and ATL2 allows the targeting of vRNA/NS5 to the endoplasmic reticulum (ER). Step 3: Viral factors and ATL2 induce the bending of the ER membrane and the formation of vesicle packets (VPs) allowing highly processive vRNA synthesis. Step 4: IGF2BP2 might be involved in the packaging of vRNA into assembling viruses by targeting the genome to the VP pore. The recruitment of IGF2BP2 to the replication compartment might be dependent on its mTOR complex 1 (mTORC1)-dependent phosphorylation status.

# Materials and methods

**Key resources table**

| Reagent type (species) or resource | Designation | Source or reference | Identifiers | Additional information |
|---|---|---|---|---|
| Gene (*Homo sapiens*) | IGF2BP2 | UniProt | Q9Y6M1 | |

*Continued on next page*

*Continued*

| Reagent type (species) or resource | Designation | Source or reference | Identifiers | Additional information |
|---|---|---|---|---|
| Strain, strain background (*Escherichia coli*) | BL21 (Rosetta DE3) | MilliporeSigma | Cat#: 70954 | Competent cells to produce proteins |
| Strain, strain background (*Escherichia coli*) | DH5α | New England Biolabs | Cat#: C2988J | Competent cells for plasmid amplification |
| Strain, strain background (orthoflavivirus) | ZIKVH/PF/2013 | European Virus Archive Global | 001v-EVA1545 | Accession:KJ776791. 2Asian lineage |
| Strain, strain background (orthoflavivirus) | ZIKV MR766 | European Virus Archive Global | 001v-EVA143 | African lineage Accession:DQ859059.1 |
| Strain, strain background (orthoflavivirus) | DENV1 HAWAII | Provided by Tom Hobman (University of Alberta, Canada) | | Serotype 1 Accession: KM204119.1 |
| Strain, strain background (orthoflavivirus) | DENV2 NGC | Provided by Tom Hobman (University of Alberta, Canada) | | Serotype 2Taxonomy ID11065 |
| Strain, strain background (orthoflavivirus) | DENV3 H87 | Provided by Tom Hobman (University of Alberta, Canada) | | Serotype 3Taxonomy ID408870 |
| Strain, strain background (orthoflavivirus) | DENV4 H241 | Provided by Tom Hobman (University of Alberta, Canada) | | Serotype 4 Taxonomy ID408686 |
| Strain, strain background (orthoflavivirus) | WNV NY99 | European Virus Archive Global | 003V-02107 | Taxonomy ID1968826 |
| Strain, strain background (betacoronavirus) | SARS-CoV-2 (Canada/QC-LSPQ-L00214517/2020) | Provided by the Public Health Laboratory of Quebec (INSPQ-LSPQ, Canada) | | GISAID: EPI_ISL_535728 |
| Genetic reagent (*Homo sapiens*) | pcDNA3-GFP-IMP2-2 | Addgene | RRID:Addgene_42175 | DNA used for IGF2BP2-HA cloning |
| Genetic reagent (*Homo sapiens*) | VCP (wt)-EGFP | Addgene | RRID:Addgene_23971 | DNA used for VCP-HA cloning |
| Cell line (*Homo sapiens*) | Huh7.5 | Provided by Patrick Labonté (INRS, Canada) | RRID:CVCL_7927 | Human hepatocarcinoma cells line, derived from Huh7 |
| Cell line (*Homo sapiens*) | NHA-hTERT | DOI:10.1038/ncomms12700 | | |
| Cell line (*Homo sapiens*) | JEG-3 | ATCC | HTB-36 | |
| Cell line (*Homo sapiens*) | Huh7-Lunet-T7 | DOI: 10.1016/j.celrep.2020.107859 | | Derived from Huh7-Lunet cells (CVCL_U459); maintained in zeocin-containing medium |
| Cell line (*Homo sapiens*) | HEK-293T | Provided by Frederick-Antoine Mallette (University of Montreal, Canada) | | |

*Continued on next page*

*Continued*

| Reagent type (species) or resource | Designation | Source or reference | Identifiers | Additional information |
|---|---|---|---|---|
| Cell line (*Cercopithecus aethiops*) | Vero E6 | ATCC | CRL-1586 | |
| Cell line (*Homo sapiens*) | HeLa | Provided by Frederick-Antoine Mallette (University of Montreal, Canada) | | |
| Cell line (*Homo sapiens*) | Huh7.5 IGF2BP2-HA | This paper | | Cell line maintained in puromycin containing medium |
| Cell line (*Homo sapiens*) | Huh7.5 VCP-HA | This paper | | Cell line maintained in puromycin containing medium |
| Transfected construct | pLKO.1-puro-shNT (plasmid) | Sigma-Aldrich | SHC002 | Lentiviral construct to produce control non-target shRNA (shNT)-expressing viruses |
| Transfected construct (*Homo sapiens*) | pLKO.1-puro-shRHA/DHX9 (plasmid) | MilliporeSigma | TRCN0000001212 | Lentiviral construct to produce shRNA-expressing viruses |
| Transfected construct (*Homo sapiens*) | pLKO.1-puro-shYBX1 (plasmid) | MilliporeSigma | TRCN0000007952 | Lentiviral construct to produce shRNA-expressing viruses |
| Transfected construct (*Homo sapiens*) | pLKO.1-puro-shDDX6 (plasmid) | MilliporeSigma | TRCN0000074696 | Lentiviral construct to produce shRNA-expressing viruses |
| Transfected construct (*Homo sapiens*) | pLKO.1-puro-shDDX21 (plasmid) | MilliporeSigma | TRCN0000051200 | Lentiviral construct to produce shRNA-expressing viruses |
| Transfected construct (*Homo sapiens*) | pLKO.1-puro-shC1QPB (plasmid) | MilliporeSigma | TRCN0000057106 | Lentiviral construct to produce shRNA-expressing viruses |
| Transfected construct (*Homo sapiens*) | pLKO.1-puro-shDDX5 (plasmid) | MilliporeSigma | TRCN0000001130 | Lentiviral construct to produce shRNA-expressing viruses |
| Transfected construct (*Homo sapiens*) | pLKO.1-puro-shYBX2 (plasmid) | MilliporeSigma | TRCN0000107507 | Lentiviral construct to produce shRNA-expressing viruses |
| Transfected construct (*Homo sapiens*) | pLKO.1-puro-shDDX3 (plasmid) | MilliporeSigma | TRCN0000000003 | Lentiviral construct to produce shRNA-expressing viruses |
| Transfected construct (*Homo sapiens*) | pLKO.1-puro-shLARP1 (plasmid) | MilliporeSigma | TRCN0000152624 | Lentiviral construct to produce shRNA-expressing viruses |
| Transfected construct (*Homo sapiens*) | pLKO.1-puro-shIGF2BP2 (plasmid) | MilliporeSigma | TRCN0000148565 | Lentiviral construct to produce shRNA-expressing viruses |
| Antibody | Anti-VCP (Mouse monoclonal) | Abcam | Cat#: ab11433 RRID:AB_298039 | WB (1:10,000) |
| Antibody | Anti-IGF2BP3 (Rabbit monoclonal) | Abcam | Cat#: ab177477 RRID:AB_2916041 | WB (1:2000) IF (1:200) |
| Antibody | Anti-YBX1 (Rabbit polyclonal) | Abcam | Cat#: ab12148 RRID:AB_2219278 | WB (1:5000) IF(1:100) |

*Continued on next page*

*Continued*

| Reagent type (species) or resource | Designation | Source or reference | Identifiers | Additional information |
|---|---|---|---|---|
| Antibody | Anti-DENV NS4B (Rabbit polyclonal) | Genetex | Cat#: GTX124250 RRID:AB_11176998 | WB (1:2000) |
| Antibody | Anti-ZIKV NS4B (Rabbit polyclonal) | Genetex | Cat#: GTX133311 RRID:AB_2728825 | WB (1:2000) |
| Antibody | Anti-ZIKV NS3 (Rabbit polyclonal) | Genetex | Cat#: GTX133309 RRID:AB_2756864 | WB (1:2000) |
| Antibody | Anti-ZIKV NS5 (Rabbit polyclonal) | Genetex | Cat#: GTX133312 RRID:AB_2750559 | WB (1:5000) |
| Antibody | Anti-ZIKV NS4A (Rabbit polyclonal) | Genetex | Cat#: GTX133704 RRID:AB_2887067 | WB (1:1000) |
| Antibody | Anti-DENV NS5 (Rabbit polyclonal) | Genetex | Cat#: GTX124253 RRID:AB_11169932 | WB (1:1000) |
| Antibody | Anti-DENV NS3 (Mouse monoclonal) | Genetex | Cat#: GTX629477 RRID:AB_2801283 | WB (1:1000) |
| Antibody | Anti-DENV2 16681 NS3 (Rat polyclonal) | MedimabsDOI:10.1111/cmi.13302 | Custom made. Previously described. | WB (1:2000) IF (1:1000) |
| Antibody | Anti-dsRNA (Mouse monoclonal) | Cedarlane | Cat#: 10010200 RRID:AB_2651015 | IF (1:100) |
| Antibody | Anti-LARP1 (Rabbit polyclonal) | Thermo Fisher Scientific | Cat#: A302-087A RRID:AB_1604274 | WB (1:2000) IF (1:100) |
| Antibody | Anti-ATL2 (Rabbit polyclonal) | Thermo Fisher Scientific | Cat#: A303-332A RRID:AB_10971492 | WB (1:1000) |
| Antibody | Anti-Actin (Mouse monoclonal) | MilliporeSigma | Cat#: A5441 RRID:AB_476744 | WB (1:10,000) |
| Antibody | Anti-HA (Mouse monoclonal) | MilliporeSigma | Cat#: H3663 RRID:AB_262051 | WB (1:5000) IF (1:1000) |
| Antibody | Anti-IGF2BP2 (Rabbit monoclonal) | Cell Signaling | Cat#: 14672S RRID:AB_2798563 | WB (1:1000) IF (1:100) |
| Antibody | Anti-IGF2BP1 (Rabbit monoclonal) | Cell Signaling | Cat#: 8482S RRID:AB_11179079 | WB (1:2000) IF (1:200) |
| Antibody | Anti-mouse, rabbit or rat Alexa Fluor (488, 568, or 647)-conjugated secondary antibodies | Thermo Fisher Scientific | Cat#: A21208 Cat#: A11029 Cat#: A-11034 Cat#: A-11031 Cat#: A-21209 Cat#: A-21247 Cat#: A-31573 Cat#: A11036 Cat#: A-21236 | Secondary antibodies used for immunofluorescence staining. Dilution: 1:10,000 |
| Other | DAPI stain | Life Technologies | D1306 | Dilution: 1:10,000 |
| Recombinant DNA reagent | pWPI | Addgene | RRID:Addgene_12254 | Lentiviral construct to transfect and express IGF2BP2-HA and VCP-HA (cloned into AscI/SpeI cassette) |
| Recombinant DNA reagent | VCP (wt)-EGFP (plasmid) | Addgene | Plasmid# 23971 RRID:Addgene_23971 | For VCP-HA cloning into pWPI via PCR |

*Continued on next page*

*Continued*

| Reagent type (species) or resource | Designation | Source or reference | Identifiers | Additional information |
|---|---|---|---|---|
| Recombinant DNA reagent | pcDNA3-GFP-IMP2-2 (plasmid) | Addgene | Plasmid# 42175 RRID:Addgene_42175 | For IGF2BP2-HA cloning into pWPI via PCR |
| Transfected construct (*Homo sapiens*) | pIRO-Z (plasmid) | DOI:10.1016/j.celrep.2020.107859 | | Transfected in Lunet-T7 cells |
| Transfected construct (*Homo sapiens*) | pFL-ZIKV-R2A (plasmid) | DOI:10.1016/j.chom.2016.05.004 | | Molecular clone to produce ZIKV-R2A (FSS13025 strain) |
| Recombinant DNA reagent | pFK-sgZIKV-R2A | DOI:10.3390/v10070368 | | Molecular clone to produce ZIKV sub-genomes |
| Recombinant DNA reagent | pFK-sgZIKV-R2A GAA | DOI:10.3390/v10070368 | | Molecular clone to produce ZIKV sub-genomes |
| Recombinant DNA reagent | pUC57 (plasmid) | Thermo Fisher | Cat#: SD0171 | Used to *in vitro* transcribe ZIKV UTR RNA |
| Sequence-based reagent | TNRC6A_F | DOI:10.1016/j.molcel.2019.11.007 | qRT-PCR primers | ACTAACTGTGGAGACCTTCACG |
| Sequence-based reagent | TNRC6A _R | DOI:10.1016/j.molcel.2019.11.007 | qRT-PCR primers | GTTAATGGGAGATGGGCTGCTA |
| Sequence-based reagent | PUM2_F | DOI:10.1016/j.molcel.2019.11.007 | qRT-PCR primers | TTTGCGCAAATACACATACGGG |
| Sequence-based reagent | PUM2 _R | DOI:10.1016/j.molcel.2019.11.007 | qRT-PCR primers | GGTCCTCCAATAGGTCCTAGGT |
| Sequence-based reagent | CIRBP_F | DOI:10.1016/j.molcel.2019.11.007 | qRT-PCR primers | GACCACGAGCCATGAGTTTTC |
| Sequence-based reagent | CIRBP _R | DOI:10.1016/j.molcel.2019.11.007 | qRT-PCR primers | CTCAGAGAAGTGAGTGGGGC |
| Sequence-based reagent | IGF2BP2_F | This paper | qRT-PCR primers | CGGGGAAGAGACGGATGATG |
| Sequence-based reagent | IGF2BP2_R | This paper | qRT-PCR primers | CGCAGCGGGAAATCAATCTG |
| Sequence-based reagent | ZIKV_F | This paper | qRT-PCR primers | AGA TGA ACT GAT TGG CCG GGC |
| Sequence-based reagent | ZIKV_R | This paper | qRT-PCR primers | AGG TCC CTT CTG TGG AAA TA |
| Sequence-based reagent | GAPDH_F | This paper | qRT-PCR primers | GAA GGT GAA GGT CGG AGT C |
| Sequence-based reagent | GAPDH_R | | qRT-PCR primers | GAA GAT GGT GAT GGG ATT TC |
| Sequence-based reagent | AscI-ATG-VCP_F | This paper | VCP cloning primers | CTGCAGGCGCGCCGCCACCATGGCTTCTGGAGCCGATTC |
| Sequence-based reagent | VCP-HA-STOP-SpeI_ R | This paper | VCP cloning primers | ACAAACTAGTTTAGTAATCAGGCACGTCATAGGGGTAACCGCCATACAGGTCATCATCA |
| Sequence-based reagent | AscI-ATG-IGF2BP2_F | This paper | IGF2BP2 cloning primers | CTGCAGGCGCGCCGCCACCATGATGAACAAGCTTTACAT |

*Continued on next page*

*Continued*

| Reagent type (species) or resource | Designation | Source or reference | Identifiers | Additional information |
|---|---|---|---|---|
| Sequence-based reagent | IGF2BP2-HA-STOP-SpeI_R | This paper | IGF2BP2 cloning primers | ACAAACTAGTTTAGTAATC AGGCACGTCATAGGGGTAA CCCTTGCTGCGCTGTGAGGCGA |
| Commercial assay or kit | mMESSAGE mMACHINE T7 transcription Kit | Thermo Fisher | Cat#: AM1344 | |
| Commercial assay or kit | Invitrogen SuperScript IV VILO Master Mix RT kit | Thermo Fisher | Cat#: 11756050 | RT-qPCR assays |
| Commercial assay or kit | Applied Biosystems SyBr Green Master mix | Thermo Fisher | Cat#: A25918 | RT-qPCR assays |
| Commercial assay or kit | ViewRNA ISH Cell Assay kit | Thermo Fisher | Cat#: QVC0001 Cat#: QVC0508 Cat#: QVC0509 Cat#: QG0507 Cat#: QVC0700 Cat#: VF4-20142 | Detection of ZIKV H/PF/2013 RNA in FISH experiments |
| Commercial assay or kit | Monoclonal Anti-HA-Agarose antibody | MilliporeSigma | Cat#: A2095 | |
| Commercial assay or kit | Trizol-LS | Thermo Fisher | Cat#: 10296010 | |
| Chemical compound, drug | NITD-008 | Tocris Small Molecules | Cat#: 6045/1 | |
| Software, algorithm | Prism 10 | GraphPad | RRID:SCR_002798 | |
| Software, algorithm | MaxQuant software v.1.6.17 | MaxQuant | RRID: SCR_014485 | |
| Software, algorithm | Perseus software v.1.6.15 | MaxQuant | RRID: SCR_015753 | |
| Software, algorithm | MO.Affinity Analysis software v.2.1.3 | NanoTemper Technologies GmbH | | |
| Software, algorithm | ImageLab software | Bio-Rad | RRID:SCR_014210 | |
| Software, algorithm | FIJI | https://imagej.net/software/fiji/ DOI:10.1038/nmeth.2019 | RRID:SCR_002285 | |

## Cells, viruses, and reagents

Human hepatocarcinoma Huh7.5 cells, Huh7-derived cells stably expressing the T7 RNA polymerase (Lunet-T7; obtained from Ralf Bartenschlager, University of Heidelberg, Germany), Vero E6 cells (CRL-1586; ATCC, Manassas, VA, USA), HEK293T cells, HeLa cells, NHA-htert (obtained from Dr. Frédérick-Antoine Mallette, University of Montréal, Canada) were all cultured in DMEM (Thermo Fisher, Burlington, Canada) supplemented with 10% fetal bovine serum (FBS; Wisent), 1% non-essential amino acids (Thermo Fisher), and 1% penicillin-streptomycin (Thermo Fisher). The identity of HEK293T and HeLa cells has been confirmed by the short tandem repeats profiling authentication method at Génome Québec (Montréal, Canada). The JEG-3 cell line (HTB-36, ATCC) was cultured in MEM with 10% FBS, 1% HEPES (Thermo Fisher), 1% sodium pyruvate (Thermo Fisher), 1% sodium bicarbonate (Thermo Fisher), and 1% penicillin-streptomycin. Huh7.5-pWPI, Huh7.5-IGF2BP2-HA, and Huh7.5-VCP-HA stable cell lines were generated by transduction of Huh7.5 with lentiviruses expressing these tagged proteins and were cultured in the presence of 1 µg/mL puromycin (Thermo Fisher). All cell lines were systematically and regularly tested for the absence of mycoplasma contamination with the PCR Mycoplasma Detection Kit (#G238 (AG); Applied Biological Mat. Inc, Richmond, Canada).

ZIKV H/PF/2013, ZIKV MR766, and WNV NY99 were provided by the European Virus Archive Global (EVAG). DENV1 HAWAII, DENV2 NGC, DENV3 H87, DENV4 H241 were kind gifts of Tom Hobman (University of Alberta, Canada). Virus stocks were amplified in Vero E6 cells following

infection at an MOI of 0.01 for 2 hr. Cell supernatants were collected at 3–7 dpi, supplemented with 1% HEPES, aliquoted and stored at 80°C until use. Infectious titers were determined by plaque assays as described before (*Freppel et al., 2018*; *Mazeaud et al., 2021*; *Sow et al., 2023*). Plasmids containing flavivirus genomes sequences encoding Rluc (pFL-ZIKV-R2A, based on FSS13025 strain), reporter sub-genomic-replicon (pFK-sgZIKV-R2A, based on H/PF/2013 strain), or replication-incompetent genomes (pFK-sgZIKV-R2A GAA) were previously reported (*Münster et al., 2018*; *Shan et al., 2016*). Plasmids were linearized using ClaI (ZIKV-R2A) or XhoI (sgZIKV-R2A and sgZIKV-R2A GAA) and subjected to *in vitro* transcription using the mMessage mMachine kit (Thermo Fisher) with T7 RNA polymerase. DENV2 16681s, ZIKV FSS13025, ZIKV-R2A particles were produced after electroporation of Vero E6 cells with *in vitro*-transcribed genomes. Vero E6 cells were resuspended in cytomix buffer (120 mM KCl, 0.15 mM $CaCl_2$, 10 mM potassium phosphate buffer [pH 7.6], 25 mM HEPES [pH 7.6], 2 mM EGTA, 5 mM $MgCl_2$ pH 7.6, freshly supplemented with 2 mM ATP, and 5 mM glutathione) at a density of $1.5 \times 10^7$ cells/mL. 10 µg of RNA were mixed with 400 µL of cells. The cells were placed in an electroporation cuvette (0.4 cm gap width; Bio-Rad, Mississauga, Canada) and pulsed with a Gene Pulser Xcell Total System (Bio-Rad) at 975 µF and 270 V. Cells were seeded in a 15 cm dish. One day after electroporation, the medium was changed. Supernatants were collected, filtered through a 0.45 µm syringe filter, and supplemented with 10 mM HEPES (pH 7.5) at 3–7 days post-electroporation. Viruses were stored at –80°C. Stocks of SARS-CoV-2 (pre-VOC isolate Canada/QC-LSPQ-L00214517/2020 [SARS-2] LSPQ (1); GISAID: EPI_ISL_535728) were produced in Vero E6 cells. The viral titers were determined by plaque assays in Vero E6 cells.

Mouse monoclonal anti-VCP (ab11433), rabbit monoclonal anti-IGF2BP3 (ab177477), and anti-YBX1 (ab12148) were purchased from Abcam (Toronto, Canada). Rabbit anti-DENV NS4B (GTX124250; cross-reactive for ZIKV), rabbit anti-ZIKV NS4B (GTX133311), rabbit anti-ZIKV NS3 (GTX133309), rabbit anti-ZIKV NS5 (GTX133312), rabbit anti-ZIKV NS4A (GTX133704), rabbit anti-DENV NS5 (GTX124253), and mouse monoclonal anti-DENV-NS3 (GTX629477; cross-reactive for ZIKV) were obtained from Genetex (Irvine, CA, USA). Rat polyclonal antibodies targeting DENV2 16681 NS3 which are cross-reactive with ZIKV NS3 were generated at Medimabs, Montréal, Canada and already reported (*Anton et al., 2021*). Mouse monoclonal anti-dsRNA (10010200) was obtained from Cedarlan. Rabbit polyclonal anti-LARP1 (A302-087A) and anti-ATL2 (A303-332A) come from Thermo Fisher Scientific. Mouse monoclonal anti-Actin (A5441) and mouse anti-HA (H3663) were purchased from MilliporeSigma (Oakville, Canada). Rabbit monoclonal anti-IGF2BP2 (14672S) and anti-IGF2BP1 (8482S) were purchased from Cell Signaling (Danvers, MA, USA).

## Plasmid design and DNA cloning

To generate tagged HA proteins IGF2BP2 and VCP expressing lentiviral constructs, PCR was performed using a plasmid VCP (wt)-EGFP gifted from Nico Dantuma (Addgene, Watertown, MA, USA; plasmid # 23971, http://n2t.net/addgene:23971; RRID:Addgene_23971) (*Tresse et al., 2010*). Forward primers expressing HA-tag fused to VCP at the C-terminus was used for this PCR. For IGF2BP2, we used pcDNA3-GFP-IMP2-2 (Addgene plasmid # 42175, https://www.addgene.org/42175/; RRID:Addgene_42175) as template and the HA-coding sequence was included in one of the primers to generate a C-terminally HA-tagged IGF2BP2. PCR products were cloned into the AscI/SpeI cassette of pWPI lentiviral plasmid.

Plasmids for the recombinant expression of IGF2BP2$_{RRM}$ and IGF2BP2$_{KH34}$ were commercially synthesized and cloned into pET28a(+) vector (Azenta Life Science, Burlington, MA, USA). The coding sequence for each of these domains was retrieved at UniProt (*Bateman et al., 2021*) using the following ID Q9Y6M1. A C-terminal hexa-histidine tag was added to IGF2BP2$_{RRM}$, whereas an N-terminal inserted into IGF2BP2$_{KH34}$. Additional plasmids were obtained for *in vitro* transcription of ZIKV 5′ and 3′ NTRs (Azenta Life Science). cDNA sequences were obtained on GenBank (Accession ID: KJ776791.2) and inserted in pUC57 plasmid flanked by an upstream T7 promoter sequence and a XbaI cutting site downstream. A pair of guanosines were added right after the promoter sequence to increase RNA yields.

Newly created DNA constructs are available freely upon request to the corresponding author.

## Lentivirus production, titration, and transduction

Sub-confluent HEK293T cells were cotransfected with pCMV-Gag-Pol and pMD2-VSV-G packaging plasmids and shRNA-encoding pLKO.1-puro plasmids or pWPI expressing HA-tagged proteins, using 25 kD linear polyethylenimine (Polysciences Inc, Warrington, PA, USA). Two- and three days post-transfection, HEK293T supernatants were collected and filtered at 0.25 µm and stored at –80°C. Lenti-viruses were titrated in HeLa cells as previously described (*Chatel-Chaix et al., 2013*). Briefly, 1 day after transduction with serially diluted lentiviruses, cells are incubated with 1 µg/mL puromycin. Five days later, cell colonies were washed twice and fixed/stained with 1% crystal violet/10% ethanol for 15–30 min. Cells were rinsed with tap water. Colonies were counted, and titers calculated considering inoculum dilution. Huh7.5 transductions were performed using an MOI of 10 in the presence of 8 µg/mL polybrene.

## Cell viability assays

MTT assays were performed to evaluate cell viability after lentiviral transduction. Huh7.5 were plated in 96-well plates (7500 cells/well) with 8 µg/mL polybrene and with the different lentiviruses at an MOI of 10. One day post-transduction, the medium was changed and after 4 days post-transduction, 20 µL of MTT at 5 mg/mL was added in the medium and incubated 1–4 hr at 37°C. Medium was removed and the MTT precipitates were dissolved with 150 µL of 2% (vol/vol) of 0.1 M glycin in DMSO (pH 11) per well. Absorbance at 570 nm was read with Spark multimode microplate reader (Tecan, Männedorf, Switzerland) with the reference at 650 nm.

## Plaque assays

$2 \cdot 10^5$ Vero E6 cells were seeded in 24-well plates. One day after plating, virus samples were seri-ally diluting to $10^{-1}$ at $10^{-6}$ fold in complete DMEM. 400 µL of serial dilutions were used to infect Vero E6 cells in duplicates (200 µL of dilution/well). 2 hr post-infection, the inoculum was removed and replaced for serum-free MEM (Thermo Fisher) containing 1.5% carboxymethylcellulose (Milli-poreSigma) for ZIKV, DENV, and WNV. For SARS-CoV-2 infected cells were incubated in MEM-0.8% carboxymethylcellulose. After 7 days for DENV2 16681s, 6 days for DENV1 HAWAII, 5 days DENV2 NGC and DENV3 H87, 4 days for ZIKV H/PF/2013, ZIKV MR766, ZIKV FSS13025, and DENV4 H241, and 3 days for WNV NY99 and SARS-CoV-2, titration are fixed during 2 hr in 2.5% formaldehyde. After thoroughly rinsing with tap water, cells were stained with 1% crystal violet/10% ethanol for 15–30 min. Stained cells were washed with tap water, plaques were counted, and titers of infectious viruses were calculated in PFU/mL.

## Rluc assay

$10^5$ Huh7.5 cells were transduced and seeded in 12-well plates. One day post-transduction, we changed the medium, and the day after, we infected them with ZIKV-R2A reporter viruses at an MOI of approximately 0.0001. 2 dpi, cells were lysed in 200 µL of luciferase lysis buffer (1% Triton X-100; 25 mM glycyl-Glycine, pH 7.8; 15 mM $MgSO_4$; 4 mM EGTA; 1 mM DTT added directly prior to use). 30 µL of lysates were plated in a 96-well white plate and luminescence was read with a Spark multi-mode microplate reader (Tecan) after injection of 150 µL of luciferase buffer (25 mM glycyl-glycine pH7.8; 15 mM $KPO_4$ buffer pH 7.8; 15 mM $MgSO_4$; 4 mM EGTA; 1 mM coelenterazine added directly prior to use). All values were background-subtracted and normalized to the control shNT-transduced cells.

## Immunofluorescence assays

Huh 7.5 cells were grown in 24-well plates containing sterile coverslips. 1 day post-seeding, cells were infected with H/PF/2013 with an MOI of 5–10. Two days post-infection, cells were washed three times with PBS before fixation, with 4% PFA for 20 min. Then, cells were washed three times with PBS before storing at 4°C. Cells were permeabilized with PBS 1X-0.2% Triton X-100 for 15 min at room temperature, and subsequently blocked for 1 hr with PBS supplemented with 5% bovine serum albumin and 10% goat serum. Cells were incubated with primary antibodies for 2 hr protected from light at room temperature. Coverslips were washed three times with PBS before incubating for 1 hr with Alexa Fluor (488, 568, or 647)-conjugated secondary antibodies, in the dark at room tempera-ture. Coverslips were then washed three times for 10 min with PBS and incubated for 15 min with

4',6'-diamidino-2-phenylindole (DAPI; Thermo Fisher, Burlington) diluted 1:10,000 in PBS. Finally, coverslips were washed three times with PBS and once with distilled water before mounting on slides with FluoromountG (SouthernBiotech, Birmingham, AL, USA). Imaging was carried out with an LSM780 confocal microscope (Carl Zeiss, Toronto, Canada) at the Confocal Microscopy Core Facility of the INRS-Centre Armand-Frappier Santé Biotechnologie and subsequently processed with the Fiji software (available at this web site: https://imagej.net/software/fiji/downloads). For the colocalization analysis, the Manders' coefficients were measured using the JacoP plugin in Fiji. In this study, the Manders' coefficient represents the fraction of NS3 or dsRNA signals overlapping with the signals of the indicated cellular RBPs.

## Co-immunoprecipitation assays

For anti-HA immunoprecipitation, cells stably expressing IGF2BP2-HA protein, or VCP-HA protein, or transduced with the control lentiviruses (pWPI) were infected with ZIKV H/PF/2013 at an MOI of 10. Two dpi, cells were washed twice with PBS, collected and lysed for 20 min on ice in a buffer containing 0.5% NP-40, 150 mM NaCl, 50 mM Tris-Cl pH 8.0, and EDTA-free protease inhibitors (Roche, Laval, Canada). The cell lysates were centrifuged at 13,000 rpm for 15 min at 4°C and supernatants were collected and stored at –80°C. Bradford assays were carried out to quantify protein concentration, before performing immunoprecipitation with the equal quantities of protein for each condition (350–800 µg of proteins) in 1–1.5 mL total volume. For *Figure 8—figure supplement 3B-C*, prior to the immunoprecipitation step, cell extracts were treated with 20 µg/mL RNase A for 20 min at room temperature. The HA immunoprecipitation was performed by incubating the lysate with 50 µL of a 50/50 slurry of mouse monoclonal anti-HA coupled to agarose beads (MilliporeSigma) for 3 hr at 4°C on a rotating wheel. The resin was washed four times with lysis buffer. The last wash was performed in new low-binding microtubes. For immunoblotting, complexes were eluted from the resin with standard SDS-PAGE loading buffer. For RNA extraction, beads were resuspended in 250 µL lysis buffer and 1 mL Trizol-LS (Thermo Fisher) was added. The samples were stored at –80°C before western blotting. The imaging of western blot membranes was done using a ChemiDoc MP System (Bio-Rad). Quantification of western blots signals was performed with the ImageLab software (Bio-Rad).

For the MS experiment, the immunoprecipitated samples were produced with the same protocol, but the final wash steps are different. After lysate incubation with the antibodies, the first three washes were done with 1 mL of lysis buffer (10 min each) at 4°C on a rotating wheel and spun down for 1 min at 10,000 rpm. The last quick wash was done with detergent-free washing buffer (150 mM NaCl, 50 mM Tris-Cl pH 8.0) and spun down for 1 min at 10,000 rpm. After the removal of buffer excess, and using a cut 200 µL tips, beads were transferred into new low-binding tubes. We performed three quick washes with 1 mL of washing buffer. After removal of the buffer, the samples were snap-frozen in ice dry and stored at –80°C until use.

## RT-qPCR

Total RNA was extracted from cells using the RNeasy Mini kit (QIAGEN, Toronto, Canada). Co-immunoprecipitated RNA was extracted from immunoprecipitated samples using Trizol-LS according to the manufacturer's instructions. RNAs were subjected to reverse transcription using the Invitrogen SuperScript IV VILO Master Mix RT kit (Thermo Fisher). Real-time PCR was performed using the Applied Biosystems SYBR green Master mix (Thermo Fisher) and a LightCycler 96 (Roche) for the detection. The following primer pairs were used: ZIKV H/PF/2013 vRNA: 5'-AGATGAACTGATGGCCGGGC-3' and 5'-AGGTCCCTTCTGTGGAAATA-3'; *GAPDH* 5'-GAAGGTGAAGGTCGGAGTC-3' and 5'-GAAGATGGTGATGGGATTTC-3'; *IGF2BP2* 5'-CGGGGAAGAGACGGATGATG-3' and 5'-CGCAGCGGGAAATCAATCTG-3'; *PUM2* 5'-TTTGCGCAAATACACATACGGG-3' and 5'-GGTCCTCCAATAGGTCCTAGGT-3'; *CIRBP* 5'-GACCACGAGCCATGAGTTTTC-3' and 5'- CTCAGAGAAGTGAGTGGGGC-3'; *TNRC6A* 5'- ACTAACTGTGGAGACCTTCACG-3' and 5'- GTTAATGGGAGATGGGCTGCTA-3'. Relative abundance of RNAs was calculated with the ΔΔCt method using *GAPDH* mRNA as a reference RNA.

## RNA FISH

Coverslip were coated with collagen I (Corning, Tewksbury, MA, USA) for 30 min and washed three times with PBS. $7 \times 10^4$ Huh 7.5 cells/well were seeded in on 24-well plates and infected 1 day later with

ZIKV H/PF/2013 with an MOI of 10. Two dpi, cells were washed three times with PBS before fixation with 4% PFA for 20 min. Then, cells were washed three times with PBS before storing at 4°C. FISH was performed using ViewRNA ISH Cell Assay kit (Thermo Fisher) according to the manufacturer's instructions. Briefly, cells were permeabilized with the kit QS detergent for 5 min, followed by three PBS washes and kit protease QS treatment. The protease QS was diluted to 1:10,000 and incubated for 10 min at room temperature. The probes for ZIKV RNA recognition (VF4-20142; Thermo Fisher) were diluted 1:100 with the kit dilution buffer. Probe hybridization was performed at 40°C for 3 hr. Subsequently, the pre-amplifying, amplifying, and fluorophore labeling steps were performed, each at 40°C for 30 min. We wash the cells like it is mentioned in the protocol. Coverslips were then immunostained with anti-IGF2BP2 antibodies and Alexa Fluor488 secondary antibodies followed by DAPI staining using a standard immunofluorescence protocol. For the colocalization analysis, the Manders' coefficient representing the fraction of vRNA signal overlapping with IGF2BP2 signal was measured using the JacoP plugin in Fiji.

## IGF2BP2 protein production for *in vitro* assays

For the expression of IGF2BP2$_{RRM}$, plasmid was transformed into Rosetta DE3 competent cells (MilliporeSigma) by heat shock. A single colony was selected and grown in 50 mL of LB media for 16–18 hr at 37°C and 220 rpm, and then transferred to a secondary culture of 500 mL of LB. Secondary cultures were incubated at 37°C and 220 rpm before induction with 1 mM IPTG when the cultures reached OD$_{600}$ reached 3.0. Following induction, cultures were incubated at 20°C and 220 rpm for 16–18 hr. Cells were centrifuged and resuspended in lysis buffer (50 mM phosphate pH 7.0, 500 mM NaCl, 10 mM imidazole). Next, we incubated the cells for 20 min at 4°C following the addition of 70 µM lysozyme, 1 µM PMSF, 80 mU/mL DNase, and 3 µM sodium deoxycholate. The cells were then lysed via sonication. Lysate was then centrifuged and passed over a HisTrap HP (Global Life Science Solutions, New York, USA) column. The column was washed with lysis buffer containing 30 mM imidazole, then eluted with a gradient elution by mixing lysis and elution buffers (50 mM phosphate pH 7.0, 500 mM NaCl, and 250 mM imidazole). The purity and homogeneity of selected samples were assessed by SDS-PAGE immediately after purification. Selected fractions were concentrated in Vivaspin 20 10 kDa MWCO centrifugal concentrators and applied to a Mono Q 10/100 GL column (Global Life Science Solutions) equilibrated with storage buffer (50 mM phosphate pH 7.0, 150 mM NaCl) to remove any residual nucleic acid. Eluted fractions were loaded in SDS-PAGE and homogenous samples were again pooled and concentrated, then passed through a Superdex 75 Increase 10/300 GL column (Global Life Science Solutions, USA) equilibrated with storage buffer as a final purification step. Samples were once again checked with SDS-PAGE, and homogenous samples were stored at 4°C until needed.

Expression and purification of IGF2BP2KH3-4 was performed as described previously (*Biswas et al., 2019*). Briefly, vectors were transformed and grown to an OD of 0.6, then induced with 1 mM IPTG. Cells were centrifuged and resuspended in lysis buffer (50 mM phosphate pH 7.0, 1.5 M NaCl, 10 mM imidazole) and then lysed through sonication. Purification was performed using nickel affinity chromatography, with eluted fractions being checked using SDS-PAGE for homogeneity. Samples were concentrated and passed over a Superdex 75 Increase 10/300 GL column as a final purification step.

## *In vitro* transcription and RNA labeling

Plasmids containing cDNA for ZIKV 5′ and 3′ NTRs were transformed and cultured in *E. coli* NEBα (New England Biolabs, Whitby, Canada) competent cells and further recovered using GeneJET Plasmid Maxiprep Kit (Thermo Fisher Scientific, Canada) as per the manufacturer's protocol, followed by a 4 hr linearization with XbaI endonuclease (New England Biolabs) at 37°C. ZIKV NTRs RNAs were synthesized by *in vitro* transcription using an in-house purified T7 polymerase, as previously described (*D'Souza et al., 2022*; *Mrozowich et al., 2023*). Subsequently, RNAs were purified on a Superdex 200 10/300 GL column (Global Life Science Solutions, USA) equilibrated with RNA buffer (10 mM Bis-Tris pH 6.0, 100 mM NaCl, 15 mM KCl, 5 mM MgCl$_2$, 5% glycerol).

Pure and homogeneous RNAs were concentrated to ~100 µM through ethanol precipitation and then fluorescently labeled at the 5′ end. The labeling reaction was then prepared from 30 µL of concentrated RNA, 1.25 mg of 1-ethyl-3-(3-dimethylamino) propyl carbodiimide hydrochloride (EDC), and 5 mg/mL of Alexa Fluor 488 dye (Thermo Fisher Scientific) dissolved in DMSO. Samples were

thoroughly mixed until contents were entirely dissolved before adding 20 µL of 0.1 M imidazole, pH 6. Afterward, we incubated the samples at room temperature for 3 hr before being further purified on a Superdex 200 10/300 GL column. RNA was checked for degradation using agarose gel electrophoresis and for labeling efficiency using MST.

## MST-based *in vitro* RNA-binding assays

MST experiments were conducted at room temperature using Nanotemper Technologies Monolith NT.115 instrument to measure binding affinity. Pure fluorescently labeled RNAs were used as targets with a fixed concentration of 250 nM for both ZIKV 5′ and 3′ NTRs. The proteins were set as the ligand and diluted in a twofold serial dilution with concentrations ranging from 65.00 to $1.98 \times 10^4$ µM for IGF2BP2$_{KH34}$ and 10.00–$6.10 \times 10^5$ µM for IGF2BP2$_{RRM}$. Reactions were prepared using MST buffer (50 mM phosphate pH 7.0, 150 mM NaCl, and 0.05% Tween 20), incubated at room temperature for 20 min and subsequently loaded into standard capillaries. The binding affinity experiments were performed using medium MST-Power, 80% excitation-power, with data collection on the cold region at 0 s and the hot region at 5 s. Three independent replicates were collected and analyzed using MO.Affinity Analysis software v.2.1.3, in which $K_d$ fitting models were obtained and plotted using GraphPad Prism 9 software.

## Affinity purification and quantitative LC-MS/MS proteomics

For the determination of IGFBP2 interactome, five independent affinity purifications using an anti-HA-conjugated agarose beads (MilliporeSigma) were performed for each experimental condition. Confluent monolayers of Huh7.5 cells constitutively expressing empty ctrls (nontarget [NT]) or HA-tagged IGFBP2 were mock-infected or infected with DENV or ZIKV at an MOI of 5 and 10, respectively. Cells were lysed in Lysis Buffer (50 mM Tris pH 7.6, 150 mM NaCl, 0.5% NP-40) containing protease and phosphatase inhibitors (cOmplete and PhosStop, Roche), and processed as previously described (*Scaturro et al., 2018*). Bound proteins (IP) or 50 µg of normalized whole cell lysates (Input) were denatured by incubation in 40 µL U/T buffer (8 M urea, 6 M thiourea, 100 mM Tris-HCl pH 8.5), and reduction and alkylation carried out with 10 mM DTT and 55 mM iodoacetamide in 50 mM ABC buffer (50 mM NH$_4$HCO$_3$ in water pH 8.0), respectively. After digestion with 1 µg LysC (Wako Chemicals, Richmond, VA, USA) at room temperature for 3 hr, the suspension was diluted in ABC buffer, and the protein solution was digested with trypsin (Promega, Madison, WI, USA) overnight at room temperature. Peptides were purified on stage tips with three C18 Empore filter discs (3M, London, Canada) and analyzed by liquid chromatography coupled to MS as previously described (*Scaturro et al., 2018*).

Samples were analyzed on a nanoElute (plugin v.1.1.0.27; Bruker, Billerica, MA, USA) coupled to a trapped ion mobility spectrometry quadrupole time of flight (timsTOF Pro) (Bruker) equipped with a CaptiveSpray source. Peptides were injected into a Trap cartridge (5 mm × 300 µm, 5 µm C18; Thermo Fisher Scientific) and next separated on a 25 cm × 75 µm analytical column, 1.6 µm C18 beads with a packed emitter tip (IonOpticks, Fitzroy, Australia). The column temperature was maintained at 50°C using an integrated column oven (Sonation GmbH, Biberach an der Riss, Germany). The column was equilibrated using four column volumes before loading samples in 100% buffer A (99.9% Milli-Q water, 0.1% formic acid [FA]). Samples were separated at 400 nL/min using a linear gradient from 2% to 17% buffer B (99.9% ACN, 0.1% FA) over 60 min before ramping up to 25% (30 min), 37% (10 min), and 95% of buffer B (10 min) and sustained for 10 min (total separation method time, 120 min). The timsTOF Pro was operated in parallel accumulation-serial fragmentation (PASEF) mode using Compass Hystar v.5.0.36.0. Settings were as follows: mass range 100–1700 m/z, 1/K0 start 0.6 V·s/cm$^2$; end 1.6 V·s/cm$^2$; ramp time 110.1 ms; lock duty cycle to 100%; capillary voltage 1600 V; dry gas 3 L/min; dry temperature 180°C. The PASEF settings were: 10 tandem MS scans (total cycle time, 1.27 s); charge range 0–5; active exclusion for 0.4 min; scheduling target intensity 10,000; intensity threshold 2500; collision-induced dissociation energy 42 eV.

## Raw data processing and analysis

Raw MS data were processed with the MaxQuant software v.1.6.17 using the built-in Andromeda search engine to search against the human proteome (UniProtKB, release 2019_10) containing forward and reverse sequences concatenated with the DENV-2 16681 strain (UniProtKB #P29990) and

ZIKV H/PF/2013 strain (UniProtKB #KU955593) with the individual viral open reading frames manually annotated, and the label-free quantitation algorithm (*Tyanova et al., 2016a*). Additionally, the intensity-based absolute quantification (iBAQ) algorithm and match between runs option were used. In MaxQuant, carbamidomethylation was set as fixed and methionine oxidation and N-acetylation as variable modifications. Search peptide tolerance was set at 70 p.p.m. and the main search was set at 30 p.p.m. (other settings left as default). Experiment type was set as TIMS-DDA with no modification to the default settings. Search results were filtered with a false discovery rate (FDR) of 0.01 for peptide and protein identification. The Perseus software v.1.6.15 was used to process the data further. Protein tables were filtered to eliminate the identifications from the reverse database and common contaminants. When analyzing the MS data, only proteins identified on the basis of at least one peptide and a minimum of three quantitation events in at least one experimental group were considered. The iBAQ protein intensity values were normalized against the median intensity of each sample (using only peptides with recorded intensity values across all samples and biological replicates) and log-transformed; missing values were filled by imputation with random numbers drawn from a normal distribution calculated for each sample. PCA was used for quality control variance across biological replicates, and remove outliers (replicates 1 and 5 of IGFBP2 pull-downs were removed from all the matrix).

Significant interactors were determined by multiple ANOVA t-tests with permutation-based FDR statistics. We performed 250 permutations, and the FDR threshold was set at 0.05. The parameter S0 was set at 1 to separate background from specifically enriched interactors. Unsupervised hierarchical clustering of proteins was performed on logarithmized and z-scored intensities of ANOVA significant interactors. Five unique clusters of proteins differentially regulated upon viral infection were identified (*Figure 9—figure supplement 1*). Results were plotted as scatter plot and heatmap using Perseus (*Tyanova et al., 2016b*) or Adobe Illustrator.

## Accession numbers and data availability

UniProtKB accession codes of all protein groups and proteins identified by MS are provided in *Supplementary file 1* and were extracted from UniProtKB (Human; release 2019_10). The MS proteomics data have been deposited to the ProteomeXchange Consortium (http://proteomecentral.proteomexchange.org) via the PRIDE partner repository with the dataset identifier PXD052835.

## Viral replication organelle induction and imaging

Lunet-T7 cells expressing a cytosolic T7 polymerase were seeded into six-well plates and transduced with lentivirus (MOI 5) containing shRNA sequences targeting either IGF2BP2 or NT. After 2 days, transduced cells were then seeded into 24-well plates (30,000 cells/well). One day later, cells were transfected with the pIRO-Z system as described in *Goellner et al., 2020*; *Cerikan et al., 2020*, using Mirus TransIT transfection reagent. After 18 hr, cells were fixed with 1.25% glutaraldehyde in 0.2 M HEPES buffer (for electron microscopy [EM] imaging) or processed for immunofluorescence, western blot, or RT-qPCR as described above.

To prepare cells for EM imaging, glutaraldehyde fixed samples were washed three times with 0.2 M HEPES. Next, samples were incubated with 1% osmium tetroxide/0.1 M cacodylate for 1 hr and washed three times with HPLC grade water. Samples were then incubated with 2% uranyl acetate for 30 min at 60°C followed by washing with HPLC-grade water three times. Progressive dehydration was performed with increasing concentrations of ethanol (40–100%). Cells were embedded into an Eponate 12 Resin (Ted Pella, Redding, CA, USA) and left for 2 days at 60°C to achieve complete polymerization. Embedded samples were sectioned into 70-nm-thick slices using a Leica EM UC6 microtome (Leica, Deerfield, IL, USA) and a diamond knife (Diatome, Quakertown, PA, USA). Sections were counterstained with 4% uranyl acetate for 5 min and 2% lead citrate for 2 min.

Sections were then mounted on EM grids for imaging. EM images were obtained on a JEOL JEM-1400 120 kV LaB6 transmission electron microscope (Jeol, Peabody, MA, USA) with a Gatan US1000 CCD camera at ×8000 magnification (Gatan, Pleasanton, CA, USA). Quantification was performed by systematically surveying cells and evaluating the presence of VPs. Only cells with >2 VPs were considered as positive. For each condition, >50 cells were surveyed over four biological replicates. All observed VPs were imaged, and VP diameters were determined using ImageJ by measuring the distance across two axes and averaging.

## Bioinformatic analyses

The gene ontology analysis was performed with ShinyGO 0.77 software (http://bioinformatics.sdstate.edu/go/). The IGF2BP2 interactomic tree was generated with STRING database using the online resource at https://string-db.org/.

## Statistical analysis

Statistical analyses were performed with GraphPad Prism 9 software. In *Figures 1 and 9* and *Figure 8—figure supplement 3*, the statistical significance was evaluated by a one-way ANOVA test (more than two test conditions compared). For *Figures 2–9 and 11*, and *Figure 9—figure supplement 2*, the statistical significance was determined using unpaired t-tests (two test conditions compared). p-Values below 0.05 were considered significant: ****: $p<0.0001$; ***: $p<0.001$; **: $p<0.01$; *$p<0.05$.

## Acknowledgements

We thank Dr. Alessia Ruggieri (University of Heidelberg) and Dr. Mirko Cortese (Telethon Institute of Genetics and Medicine) for excellent scientific discussion about this project. We are grateful to Jessy Tremblay and Arnaldo Nakamura at the Confocal Microscopy and Flow Cytometry Facility and the Electron Microscopy Facility of INRS-Centre Armand-Frappier, respectively, for excellent technical assistance. The authors wish to acknowledge the support of the Emory University Robert P Apkarian Integrated Electron Microscopy Core Facility (RRID: SCR_023537) for help with preparing EM samples and with establishing imaging workflows. We thank Dr. Pei-Yong Shi, the World Reference Center for Emerging Viruses and Arboviruses (WRCEVA) and Dr. Ralf Bartenschlager (University of Heidelberg) for providing the ZIKV and DENV reverse genetics systems, and Dr. Daniel Lamarre, Dr. Frédérick Antoine Mallette (University of Montréal), Dr. Tom Hobman (University of Alberta), Dr. Patrick Labonté (Institut National de la Recherche Scientifique), and Dr. Anil Kumar (University of Saskatchewan) for generously providing us with shRNA-expressing lentiviral plasmids and cell lines. We are grateful to the European Virus Archive Global (EVAg), Dr. Xavier de Lamballerie (Emergence des Pathologies Virales, Aix-Marseille University, France) and Robin Gafur (Animal and Plant Health Agency, Addlestone, UK) for providing ZIKV H/PF/2013 and WNV99 original stocks. We thank the Laboratoire de Santé Publique du Québec and Philippe Dufresne for providing the SARS-CoV-2 isolate, and Dr. Alain Lamarre and Mrs Tania Charpentier for help with the amplification of this virus. CM and AAS received PhD fellowships from the Armand-Frappier Foundation and the Center of Excellence in Research on Orphan Diseases-Courtois Foundation (CERMO-FC). AAS is receiving a PhD fellowship from the Fonds de Recherche du Québec-Nature et Technologies (FRQNT). HSP is receiving a CREATE Postdoctoral Fellowship award from the Natural Sciences and Engineering Research Council of Canada (NSERC). TRP is supported by a NSERC Discovery grant (RGPIN-2022–03391) and the Canada Foundation for Innovation (CFI-37155 and CFI-41008), and acknowledges the Canada Research Chair program. LCC is receiving a senior research scholar salary support from Fonds de Recherche du Québec-Santé (FRQS). Work in PS laboratory is funded by the Free and Hanseatic City of Hamburg, the German Federal Ministry of Education and Research (VirMScan) and the German Research Foundation (SC 314/1-2). This project was supported by a Discovery grant from NSERC (RGPIN-2016-05584) and an Early Career Investigator grant from FRQNT (2018-NC-205593), awarded to LCC.

## Additional information

### Funding

| Funder | Grant reference number | Author |
|---|---|---|
| Natural Sciences and Engineering Research Council of Canada | RGPIN-2022-03391 | Trushar R Patel |
| Natural Sciences and Engineering Research Council of Canada | RGPIN-2016-05584 | Laurent Chatel-Chaix |

| Funder | Grant reference number | Author |
|---|---|---|
| Canada Foundation for Innovation | CFI-37155 | Trushar R Patel |
| Canada Foundation for Innovation | CFI-41008 | Trushar R Patel |
| Fonds de recherche du Québec – Nature et technologies | 2018-NC-205593 | Laurent Chatel-Chaix |
| Bundesministerium für Bildung und Forschung | VirMScan | Pietro Scaturro |
| Deutsche Forschungsgemeinschaft | SC 314/1-2 | Pietro Scaturro |
| Free and Hanseatic City of Hamburg | | Pietro Scaturro |
| Fonds de recherche du Québec – Nature et technologies | PhD fellowship | Aïssatou Aïcha Sow |
| Armand-Frappier Foundation | PhD fellowship | Clément Mazeaud Aïssatou Aïcha Sow |
| Center of Excellence in Research on Orphan Diseases – Fondation Courtois | PhD fellowship | Clément Mazeaud Aïssatou Aïcha Sow |
| Natural Sciences and Engineering Research Council of Canada | CREATE postdoctoral fellowship award | Higor Sette Pereira |
| Canada Research Chairs | | Laurent Chatel-Chaix Trushar R Patel |

The funders had no role in study design, data collection and interpretation, or the decision to submit the work for publication.

### Author contributions

Clément Mazeaud, Formal analysis, Investigation, Methodology, Writing - original draft, Writing – review and editing; Stefan Pfister, Jonathan E Owen, Higor Sette Pereira, Flavie Charbonneau, Zachary E Robinson, Cheyanne L Bemis, Aïssatou Aïcha Sow, Formal analysis, Investigation, Methodology; Anaïs Anton, Investigation, Methodology; Trushar R Patel, Resources, Investigation, Writing – review and editing; Christopher J Neufeldt, Resources, Formal analysis, Investigation, Methodology, Writing – review and editing; Pietro Scaturro, Resources, Software, Formal analysis, Investigation, Methodology, Writing – review and editing; Laurent Chatel-Chaix, Conceptualization, Resources, Formal analysis, Supervision, Funding acquisition, Project administration, Writing – review and editing

### Author ORCIDs

Clément Mazeaud http://orcid.org/0009-0003-0152-4615
Higor Sette Pereira https://orcid.org/0000-0003-4761-8361
Trushar R Patel https://orcid.org/0000-0003-0627-2923
Christopher J Neufeldt https://orcid.org/0000-0002-4551-1811
Laurent Chatel-Chaix https://orcid.org/0000-0002-7390-8250

Reviewer #1 (Public Review): https://doi.org/10.7554/eLife.94347.3.sa1
Reviewer #2 (Public Review): https://doi.org/10.7554/eLife.94347.3.sa2
Reviewer #3 (Public Review): https://doi.org/10.7554/eLife.94347.3.sa3
Author response https://doi.org/10.7554/eLife.94347.3.sa4

## Additional files

### Supplementary files

• Supplementary file 1. Quantitative mass spectrometry-based analysis of changes in the IGF2BP2 interactome during Zika virus (ZIKV) and dengue virus (DENV) infections. This searchable spreadsheet indicates the specific IGF2BP2 protein partners whose association is decreased or increased during ZIKV and/or DENV infection, or remained unchanged. Hits for each category are indicated with a (+). All conditions were compared to the uninfected IGF2BP2-HA condition. Protein LFQ intensity values for each biological replicate are also included in the table. This spreadsheet relates to *Figure 9*.

• MDAR checklist

### Data availability

All data generated or analysed during this study are included in the manuscript and supporting files; source data files have been provided for Figures 1-8, 10 and all Figure supplements (except Fig 9 Supplement 1 whose data are in Suppl File 1 and have been deposited online). The mass spectrometry proteomics data have been deposited to the ProteomeXchange Consortium via the PRIDE partner repository with the dataset identifier PXD052835.

The following dataset was generated:

| Author(s) | Year | Dataset title | Dataset URL | Database and Identifier |
|---|---|---|---|---|
| Mazeaud C, Pfister S, Scaturro P, Chatel-Chaix L | 2024 | Zika virus remodels the IGF2BP2 and VCP cellular interactome | https://www.ebi.ac.uk/pride/archive/projects/PXD052835 | PRIDE, PXD052835 |

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
