## [Editor Report · eLife assessment]

This **valuable** study combines multidisciplinary approaches to examine the role of insulin-like growth factor 2 mRNA-binding protein 2 (IGF2BP2) as a potential novel host dependency factor for Zika virus. The main claims are supported by the data but remain **incomplete**. The evidence would be strengthened by improving the western blot analyses and adjusting the toning of their claims in relation to the role of IGF2BP2 for viral replication. With the experimental evidence strengthened, this work will be of interest to virologists working on flaviviruses.

---

## [Referee Report · Reviewer #1 (Public Review)]

Summary:

This study investigated the co-option of IGF2BP2, an RNA binding protein by ZIKV proteins. Designed experiments evaluated if IFG2BP2 co-localized to sites of viral RNA replication, interacted with ZIKV proteins and how ZIKV infection changed the IGF2BP2 interactome.

Strengths:

The authors have used multiple interdisciplinary techniques to address several questions regarding the interaction of ZIKV proteins and IGF2BP2.

The findings could be exciting if concerns are addressed, specifically regarding how ZIKV infection alters the interactome of IGF2BP2.

Comments on thee revised version:

Following response to reviews, the authors have addressed a majority of the concerns with the exception of the western blots:

As requested in the previous review, the authors did quantify the western blot data for half of the blot in 2A, but did not quantify blots in D and E. Please quantify ALL blots. Also, the first two lanes of 2A. The same goes for 4A only infected is quantified, please quantify Mock as well. In the quantification of 4C, all lanes should be quantified, not only the NS5 from C. Also, unclear which lanes were quantified (H/PF/2013 or MR766)? Also, quantification needs to be generally shown as a graph and not included on top of the western blot.

---

## [Referee Report · Reviewer #2 (Public Review)]

Clément Mazeaud et al. identified the insulin-like growth factor 2 mRNA-binding protein 2 (IGF2BP2) as a proviral cellular protein that regulates Zika virus (ZIKV) RNA replication by modulating the biogenesis of virus-induced replication organelles. Based on their findings and previously published data, the authors propose a model outlining the role of IGF2BP2 in the ZIKV infectious cycle. This model details the changes in IGF2BP2 interactions with both cellular and viral proteins and RNAs during viral infection.

Strengths:

This revised manuscript presents an interesting and convincing mechanism by which a cellular RNA-binding protein alters its protein and RNA interactome during viral infection. Using various molecular biology methods, proteomic analysis and a newly described replication-independent vesicle packets induction system, the authors describe the relevance of IGF2BP2 protein during Zika virus infection.

Weaknesses:

In the proposed model, the IGF2BP2 protein specifically binds to the 3' nontranslated region (NTR) of the ZIKV genome, while excluding binding to the 5' NTR. However, the authors cannot rule out the possibility that this host protein associates with other regions of the viral genome, a topic which is discussed in the manuscript.

In this study, the physiological cellular consequences of altering the interaction of IGF2BP2 with its endogenous mRNA ligands due to ZIKV infection remain unexplored. This aspect would be of interest for future studies.

---

## [Referee Report · Reviewer #3 (Public Review)]

Summary:

The manuscript by Mazeaud and colleagues pursued a small scale screen of a targeted RNAi library to identify novel players involved in Zika (ZIKV) and dengue (DENV) virus replication. Loss-of-function of IGF2BP2 resulted in reduced titers for ZIKV of the Asian and African lineages in hepatic Huh7.5 cells, but not for either of the four DENV serotypes nor for West Nile virus (WNV). The phenotype was further confirmed in two additional cell lines and using a ZIKV reporter virus. In addition, using immunoprecipitation assays the interaction between IGF2BP2 and ZIKV NS5 protein and RNA genome was detected. The work addressed the role of IGF2BP2 in the infected cell combining confocal microscopy imaging, and proteomic analysis. The approach indicated an altered distribution of IGF2BP2 in infected cells and changes in the protein interactome including disrupted association with partner mRNAs and modulation of the abundance of a specific set of protein partners in IGF2BP2 immunoprecipitated ribonucleoprotein (RNP) complexes. Finally, based on the changes in IGF2BP2 interactome and specifically the increment in the abundance of Atlastin 2, biogenesis of ZIKV replication organelles (vRO) is investigated using a genetic system that allows virus replication-independent assembly of vRO. Electron microscopy showed that knock down of IGF2BP2 expression reduced the number of cells with vRO.

Strengths:

The role of IGF2BP2 as a proviral factor for ZIKV replication is novel

The study follows a logical flow of experiments that altogether support the assembly of a specialized RNP complex containing IGF2BP2 and ZIKV NS5 and RNA genome

Weaknesses:

The specificity for the direct interaction between IGF2BP2 and ZIKV RNA genome remains elusive in particular regarding the regions in the virus genome that drive interaction.

---

## [Author Response]

The following is the authors’ response to the original reviews.

This valuable study combines multidisciplinary approaches to examine the role of insulin-like growth factor 2 mRNA-binding protein 2 (IGF2BP2) as a potential novel host dependency factor for Zika virus. The main claims are partially supported by the data, but remain incomplete. The evidence would be strengthened by improving the immunofluorescence analyses, addressing the role of IGF2BP2 in "milder" infections, and elucidating the role of IGF2BP2 in the biogenesis of the viral replication organelle. With the experimental evidence strengthened, this work will be of interest to virologists working on flaviviruses.

We thank the reviewers for their feedback and constructive suggestions. In this revised version of the manuscript, we have addressed the reviewer’s comments to the best of our ability as detailed below. We believe that the newly incorporated data strengthens our study and conclusions. We hope that this revised manuscript will satisfy the reviewers and will be of high interest to flavivirologists.

**Public Reviews:**

**Reviewer #1 (Public Review):**
Summary:This study investigated the co-option of IGF2BP2, an RNA-binding protein by ZIKV proteins. Designed experiments evaluated if IFG2BP2 co-localized to sites of viral RNA replication, interacted with ZIKV proteins, and how ZIKV infection changed the IGF2BP2 interactome.Strengths:The authors have used multiple interdisciplinary techniques to address several questions regarding the interaction of ZIKV proteins and IGF2BP2.The findings could be exciting, specifically regarding how ZIKV infection alters the interactome of IGF2BP2.

We thank the reviewer for acknowledging the multidisciplinary approach of our study and its exciting potential.

Weaknesses:Significant concerns regarding the current state of the figures, descriptions in the figure legends, and the quality of the immunofluorescence and electron microscopy exist.

In this new version of the manuscript, we have improved the quality of the microscopy data and included the requested information in the figure legends as described below in the Recommendations section.

**Reviewer #2 (Public Review):**
Clément Mazeaud et al. identified the insulin-like growth factor 2 mRNA-binding protein 2 (IGF2BP2) as a proviral cellular protein that regulates Zika virus RNA replication by modulating the biogenesis of virus-induced replication organelles.The absence of IGF2BP2 specifically dampens ZIKV replication without having a major impact on DENV replication. The authors show that ZIKV infection changes IGF2BP2 cellular distribution, which relocates to the perinuclear viral replication compartment. These assays were conducted by infecting cells with an MOI of 10 for 48 hours. Considering the ZIKV life cycle, it is noteworthy that at this time there may be a cytopathic effect. One point of concern arises regarding how the authors can ascertain that the observed change in localization is a consequence of the infection rather than of the cytopathic effect. To address this concern, shorter infection periods (e.g., 24 hours post-infection) or additional controls, such as assessing cellular proteins that do not change their localization or infecting with another flavivirus lacking the IGF2BP2 effect, could be incorporated into their experiments.

We thank the reviewer for these relevant comments regarding the specificity of IGF2BP2 relocalization to the ZIKV replication compartment.

It is noteworthy that we chose the 2-day post-infection time point for our analyses because it corresponds to the peak of replication with much more titers produced compared to those at 24 hours post-infection (generally ~106 PFU/mL vs. ~104 PFU/mL). Consistently, the abundance of viral replication factories is more obvious at this time-point. A MOI of 5-10 was chosen to maximize the % of infected cells. That said, as suggested by the reviewer, we have analyzed the distribution of IGF2BP2 in ZIKV-infected cells at one-day post-infection, and we provide evidence in Figure S1 that IGF2BP2 relocalizes to the dsRNA-containing compartment at this time point.

Importantly, we now show in Figure S5 that in contrast to IGF2BP2, other host RNA-binding proteins such as LARP1 and DDX5 do not accumulate to ZIKV replication compartment at 2 days post-infection. LARP1 actually seems to be excluded from it while DDX5 remains nuclear. Of note, consistent with the ZIKV-induced decrease in expression observed in western blots (Fig 4A), the intensity of DDX5 signal decreases in infected cells. Altogether, this demonstrates that the IGF2BP2 relocalization phenotype is specific and is not due to ZIKV-induced cell death.

By performing co-immunoprecipitation assays on mock and infected cells that express HAtagged IGF2BP2, the authors propose that the observed change in IGF2BP2 localization results from its recruitment to the replication compartment by the viral NS5 polymerase and associated with the viral RNA. Given that both IGF2BP2 and NS5 are RNA-binding proteins, it is plausible that their interaction is mediated indirectly through the RNA molecule. Notably, the authors do not address the treatment of lysates with RNase before the IP assay, leaving open the possibility of this indirect interaction between IGF2BP2 and NS5.

We agree with the hypothesis of the reviewer. As suggested, we have performed coimmunoprecipitation assays following RNase A treatment of the cell lysates. As shown in new Fig S6, the abundance of ZIKV NS5 co-immunoprecipitating with IGF2BP2-HA is drastically decreased upon RNase A treatment compared to the untreated condition. This demonstrates that the IGF2BP2/NS5 interaction is mostly RNA-dependent, which is not surprising as RNA is often a structural component of ribonucleoprotein complexes. Of note, the same is observed with ATL2. This new set of data allows us to refine our model of Figure 11 and the discussion as they strongly suggest that the direct binding of IGF2BP2 to viral RNA (evidenced in vitro; Fig 5D) is required for subsequent association with NS5 and ER-shaping protein ATL2. This is in line with the fact that viral RNA is a co-factor in the biogenesis of ER-derived ZIKV vesicle packets (PMID: 32640225). However, we cannot exclude a contribution of cellular RNA in these processes as discussed.

In in vitro binding assays, the authors demonstrate that the RNA-recognition motifs of the IGF2BP2 protein specifically bind to the 3' nontranslated region (NTR) of the ZIKV genome, excluding binding to the 5' NTR. However, they cannot rule out the possibility of this host protein associating with other regions of the viral genome. Using a reporter ZIKV subgenomic replicon system in IGF2BP2 knock-down cells, they additionally demonstrate that IGF2BP2 enhances viral genome replication. Despite its proviral function, the authors note that the "overexpression of IGF2BP2 had no impact on total vRNA levels." However, the authors do not delve into a discussion of this latter statement.

We agree with the reviewer’s comments. We now mention in the discussion that we cannot exclude the possibility that IGF2BP2 associates with RNA motifs within the coding region of the viral genomic RNA, especially considering that it contains N6A-methylated sequences (PMID: 27773535; 27773536; 29373715). Moreover, we discuss the observation that IGF2BP2 overexpression has no impact on vRNA levels (as well as titers). We believe that this is because endogenous IGF2BP2 is highly expressed in cancer cells such as the Huh7.5 and JEG-3 cells used here and is presumably not limiting for viral replication in our system (PMID: 38320625; 35111811; 34309973; 35023719; 37088822; 33224879; 35915142).

In this study, the authors extend their findings by illustrating that ZIKV infection triggers a remodeling of IGF2BP2 ribonucleoprotein complex. They initially evaluate the impact of ZIKV infection on IGF2BP2's interaction with its endogenous mRNA ligands. Their results reveal that viral infection alters the binding of specific mRNA ligands, yet the physiological consequences of this loss of binding in the cell remain unexplored.

We acknowledge that it would be of interest to further study the physiological relevance of the modulation of IGF2BP2 ribo-interactome. Since we have focused here on the role of IGF2BP2 in viral replication, we feel that this will be the focus of future studies notably involving a larger omic-centered approach to identify the most impacted IGF2BP2 mRNA ligands. Of note, Gokhale and colleagues have already reported that CIRBP, *TNRC6A* and *PUM2* proteins regulates the replication of *Flaviviridae* (PMID: 31810760).

Additionally, the authors demonstrate that ZIKV infection modifies the IGF2BP2 interactome. Through proteomic assays, they identified 62 altered partners of IGF2BP2 following ZIKV infection, with proteins associated with mRNA splicing and ribosome biogenesis being the most represented. In particular, the authors focused their research on the heightened interaction between IGF2BP2 and Atlastin 2, an ER-shaping protein reported to be involved in flavivirus vesicle packet formation. The validation of this interaction by Western blot assays prompted an analysis of the effect of ZIKV on organelle biogenesis using a newly described replication-independent vesicle packet induction system. Consequently, the authors demonstrate that IGF2BP2 plays a regulatory role in the biogenesis of ZIKV replication organelles.Based on these findings and previously published data, the authors propose a model outlining the role of IGF2BP2 in ZIKV infectious cycle, detailing the changes in IGF2BP2 interactions with both cellular and viral proteins and RNAs that occur during viral infection.The conclusions drawn in this paper are generally well substantiated by the data.

We thank the reviewers for this encouraging general comments on our study.

However, it is worth noting that the majority of infections were conducted at a high MOI for 48 hours, spanning more than one infectious cycle. To enhance the robustness of their findings and mitigate potential cell stress, it would be valuable to observe these effects at shorter time intervals, such as 24 hours post-infection.

As explained above, IGF2BP2 relocalization to the (dsRNA-enriched) replication compartment was also observed in ZIKV infected cells at one day post-infection.

Furthermore, the assertion regarding the association of IGF2BP2 with NS5 could be strengthened through additional immunoprecipitation (IP) assays. These assays, performed in the presence of RNAse treatment, would help exclude the possibility of an indirect interaction between IGF2BP2 and NS5 (both RNA-binding proteins) through viral RNA, thus providing more confidence in the observed association.

See above for our answer and the description of the new data of Fig. S7.

**Reviewer #3 (Public Review):**
Summary:The manuscript by Mazeaud and colleagues pursued a small-scale screen of a targeted RNAi library to identify novel players involved in Zika (ZIKV) and dengue (DENV) virus replication. Loss-of-function of IGF2BP2 resulted in reduced titers for ZIKV of the Asian and African lineages in hepatic Huh7.5 cells, but not for either of the four DENV serotypes nor West Nile virus (WNV). The phenotype was further confirmed in two additional cell lines and using a ZIKV reporter virus. In addition, using immunoprecipitation assays the interaction between IGF2BP2 and ZIKV NS5 protein and RNA genome was detected. The work addressed the role of IGF2BP2 in the infected cell combining confocal microscopy imaging, and proteomic analysis. The approach indicated an altered distribution of IGF2BP2 in infected cells and changes in the protein interactome including disrupted association with partner mRNAs and modulation of the abundance of a specific set of protein partners in IGF2BP2 immunoprecipitated ribonucleoprotein (RNP) complexes. Finally, based on the changes in IGF2BP2 interactome and specifically the increment in the abundance of Atlastin 2, the biogenesis of ZIKV replication organelles (vRO) is investigated using a genetic system that allows virus replication-independent assembly of vRO. Electron microscopy showed that knockdown of IGF2BP2 expression reduced the number of cells with vRO.Strengths:The role of IGF2BP2 as a proviral factor for ZIKV replication is novel. The study follows a logical flow of experiments that altogether support the assembly of a specialized RNP complex containing IGF2BP2 and ZIKV NS5 and RNA genome.

We thank the reviewer for their positive feedback on our study and its novelty.

Weaknesses:The statistical analysis should clearly indicate the number of biological replicates of experiments to support statistical significance.

This information has been included in all figure legends.

The claim that IGF2BP2 knockdown impairs *de novo* viral organelle biogenesis and viral RNA synthesis is built upon data that show a reduction in RNA synthesis <0.5-fold as assessed using a reporter replicon, thus suggesting a limited impact of the knockdown on RNA replication.

We agree that a 50% decrease in the replication of our reporter replicon might be considered mild. However, we want to pinpoint that in an infectious set-up, the phenotypes were higher as demonstrated by an 80% decrease in viral particle production even when IGF2BP2 levels were never depleted more that 80% compared to endogenous levels. Moreover, our findings were validated through the analysis of *de novo* vRO biogenesis by electron microscopy in a replication-independent set-up. Together, these experiments provide compelling evidence for a role for IGF2BP2 in the early stages of viral genome replication.

Validation of IGF2BP2 partners that are modulated upon ZIKV infection (i.e. virus yield in knocked down cells) can be relevant especially for partners such as Atlastin 2, as the hypothesis of a role for IGF2BP2 RNP in vRO biogenesis is based on the observed increase in the abundance of Atlastin 2 in the RNP complex preciìtated from infected cells.

First, we would like to emphasize that the proviral role of ATL2 in flavivirus replication, including links to vRO biogenesis, was already reported in two independent studies notably by one of the co-authors (PMID: 31636417; 31534046). Therefore, we have chosen to discuss these previous studies in the manuscript rather than repeating published experiments. Second, we agree that it would be interesting to further interrogate the role of modulated IGF2BP2 protein partners in ZIKV replication. However, these experiments would constitute a new project per se involving fastidious RNAi-based phenotypic screening and subsequent functional characterization of the identified hits. Therefore, this will be the focus of follow-up studies.

**Recommendations for the Authors:**

**Reviewer #1 (Recommendations For The Authors):**
All IFAs claimed that showing co-localization is minimal, this needs to be addressed.

We have performed colocalization analyses for relevant images in the revised manuscript see below and Figs. 4B, 5A, S4A-C and S5A-D. Although this quantification increases confidence in our analysis, we were still cautious in our conclusions, stating that colocalization was partial and that IGF2BP2 accumulates in the replication compartment.

Western blots and IPs need to be quantified.

As requested, we have included WB quantification in Figs. 2A, 4A, 4D, 8B-D, S6C and S7D.

Figure 1: What is the strain background for the ZIKV reporter virus?

As indicated in the legend of Figure 1E of the primary submission, the Rluc-expressing ZIKV reporter virus (ZIKV-R2A) was based on the FSS13025 isolate (Asian lineage)(PMID: 27198478). To clarify this, we have also indicated the strain background in the main text of the Results and Material & Methods sections.

Figure 2A: If shGF2BP2 reduces viral titer, the NS3 should show a reduction in 2A, but it doesn't.

We agree with the reviewer. Although NS3 seems not to be decreased upon IGF2BP2 knockdown in the experiment initially shown in Figure 2A, it should be noted that our homemade rat anti-NS3 antibody is highly sensitive, leading to signal saturation that makes it challenging to distinguish changes in NS3 expression without diluting substantially the lysate sample before the PAGE-SDS. The initial reason for including Fig 2A was not to make a statement about viral protein expression but to validate IGF2BP2 knock-down efficiency. Conclusions about NS3 levels in the initial figure are further complicated by the high MOI of ZIKV was used in Huh7.5 cells which are not quantitative for viral replication measurements. To address this issue, we assessed the impact of IGF2BP2 knockdown on viral protein abundance (as a read-out of overall viral replication) with a lower MOI of ZIKV. The results of the repeat experiment (seen in the new Fig. 2A) show that IGF2BP2 knockdown leads to a decrease in the abundance of NS4A, NS5 and NS3, which is consistent with the titer decrease phenotypes.

Figure S3: The re-localization claimed is minimal and does not show overlap with NS3. The dsRNA is difficult to see here. Suggest improving the immunofluorescence images and reducing the claim for "strong" co-option of RNP complexes.

In addition to replication complexes, NS3 labels convoluted membranes which are devoid of dsRNA and IGF2BP2 and surround the cage-like replication compartment as large puncta (PMID: 27545046; 33432690; 28249158). The signal overlap is more obvious between IGF2BP2 and NS3/dsRNA-containing areas, which is reflected by the Mander’s coefficients that have been included in the revised version (Fig. S5C-D). We have also adjusted the text to conclude that the colocalization was partial and that IGF2BP2 accumulated in the replication compartment. We acknowledge that the dsRNA signal is weak, and we have updated the images (and others, when relevant) to better visualize this viral component. Moreover, we have rephrased the sentence to remove the word “strongly”.

Figure 4A: Western blot needs quantification.

This is now included in the figure.

Figure 4B: As in many of the IFAs, the co-localization is only partial. Additionally, the dsRNA is not visible. So the images need to be improved. The colocalization should be quantified across the cell diameter.

We changed the color and intensity of the dsRNA staining to make it more visible. Mander’s colocalization coefficients have been determined and included in Figures 4B and S5C-D.

Figure 4C: It is difficult to understand what the +/- is on the blots for the cell extracts and the anti-HA IP samples. It is not described in the figure legend or the text.

As already indicated on the right of the panel, the +/- indicates whether or not IGF2BP2-HA was overexpressed in the cells. In the revised version, this is clarified in the figure legend.

Figure 5A: Once again similar to other IFAs, the co-localization is only minimal and thus difficult to claim as "co-localization" is actually happening. It would be good to either improve the images or discuss this observation in the text and reduce the claim of colocalization. Specifically, since the two proteins might be co-localizing in specific regions which would make it a very interesting observation. Also, quantification of co-localizing regions would be beneficial.

We have included the requested colocalization analysis. We have been cautious to indicate that colocalization was only partial. It is noteworthy that, despite many efforts in the optimization of the cell permeabilization procedure, we noticed that the FISH probes were not very efficient in accessing the perinuclear area of the infected cells, where replication complexes accumulate. In that respect, it is likely that this imaging approach “miss” some of the IGF2BP2/vRNA complexes and that the determined colocalization factor is underestimated. This explains why the confirmation of the vRNA/IGF2BP2 complex with a biochemical approach (Fig. 5B) was very relevant.

Figure 5D: It is unclear what the blue squares represent. Clearer figure legends and text would be beneficial.

As stated in the initial figure, the blue squares indicate values obtained with the ZIKV 5’ UTR probe while the green circles involve a 3’ UTR probe. We have further emphasized this information in the figure legend to make it clearer.

Figure 6B. The graph is missing the data and X-axis label for shIGF2BP2.

We had initially omitted the values of the conditions with shIGF2BP2 and the replicationdead GAA replicon, since this viral system does not allow accumulation of viral genomes or proteins and was not relevant at the 48h time point. We thought that the inclusion of the shNT/GAA condition was enough an internal negative control of viral replication since values for shIGF2BP2/GAA did not exceed background. Nevertheless, we have now included this condition in the revised figure.

Figure 7D: It is unclear what the -/+ signs are in the cell extracts and the IP blots. Specifically, since there is an NS5 signal in the (-) lanes.

As explained above, the +/- indicates whether IGF2BP2-HA was overexpressed. The meaning of these symbols is now further clarified in the figure legend.

Figure 8C: The circles with the different colors are not clearly described. What does it mean?

As indicated in the figure (left part), the red and green circles identify the partners of the STRING network whose association with IGF2BP2 is decreased and increased during infection, respectively. We have included this information in the figure legend.

Figure 9: The electron microscopy to quantify vesicles should be carried out using whole-cell tomography in order to get the most accurate quantification of the vesicles following different treatments. This is because if you only look at one cell profile (slice), the number of vesicles might be less in that profile and more in another below or above it. It is unclear how many cell profiles were used for the quantification and how the calculations were carried out.

We agree with the reviewer that ideally, one should perform 3D electron tomography to precisely assess the morphology of VPs. Regardless the fact that we do not possess the imaging infrastructure to perform that type of analysis, such an approach would represent a tremendous amount of work if one would like to process at least 200-400 vesicles from > 50 cells and their whole cytoplasm (as we did). Despite not having 3D images, this number of data points is sufficient to see general changes in viral replication vesicle morphology, especially considering that Huh7-Lunet cells are relatively flat cells. (PMID: 32640225; 36700643; 34696522; 31636417). Furthermore, since IGF2BP2 knockdown decreases the abundance of VPs and does not impact their diameter, we believe that the addition of sophisticated 3D analysis would not bring any new and relevant information and that the TEM data stand by themselves for the conclusion we made. A more refined morphological analysis to determine how IGF2BP2 is structurally involved in virus-mediated membrane reorganization could be the focus of a future study.

We feel that we have already provided sufficient information about the quantification in the Material & Methods section of the first version of the manuscript: “Quantification was performed by systematically surveying cells and evaluating the presence of VPs. Only cells with >2 VPs were considered as positive. For each condition, >50 cells were surveyed over 4 biological replicas. All observed VPs were imaged, and VP diameters were determined using ImageJ by measuring the distance across two axes and averaging”.

**Reviewer #2 (Recommendations For The Authors):**
The inclusion of a control in the knock-down and infection assays with the reporter virus could enhance the validity of the findings. Introducing STAT2 knockdown, a recognized antiviral protein for ZIKV, as a control would provide a valuable benchmark to evaluate the extent of viral enhancement in the experiments. This additional control not only supports the proposed function of LARP1 in virus assembly/release but also strengthens the overall interpretation of the results.

We agree that adding a positive control could have been relevant for assessing the extent of replication modulation, especially for increases such as that observed with shLARP1. However, finding such control proteins in our system was a challenge. Indeed, STAT2 would not have been a good control for these experiments since we used Huh7.5 cells for the RNAi mini-screening, which do not express a functional RIG-I protein, and generally do not produce type I and III interferons. Thus, STAT2 knockdown is not expected to result in an increase in replication. That said, we feel that it was unnecessary to include a control for replication inhibition here given that only a few statistically reliable candidates we obtained. Instead, we have opted for an extensive secondary validation approach by assessing the proviral role of IGF2BP2 for multiple viruses - DENV1-2-3-4, WNV and SARS-CoV-2, and 3 ZIKV strains in three relevant cell types.

Additionally, in Figure S4, the authors employ an antibody against NS5 that specifically recognizes ZIKV NS5 but not DENV NS5. Given the objective of highlighting distinctions between these two viruses, it is advisable to use an antibody that detects DENV NS5 as well. This approach would contribute to a more comprehensive comparison, ensuring a balanced representation of both viruses in the experimental analysis.

We thank the reviewer for this relevant suggestion. We have repeated the coimmunoprecipitation assays using antibodies specific to DENV NS5 (Aithor response image 1). While we specifically pulled down ZIKV NS5 with IGF2BP2-HA as expected, this was not the case for DENV NS5 when using extracts from DENV-infected cells despite our multiple attempts. Indeed, the amount of pulled-down DENV NS5 with IGF2BP2-HA was always comparable to that in the negative control (“empty” pWPI lentivirus-transduced cells, “-“ condition), which corresponds to non-specific binding to the HA-resin. Thus, while the antibody was very efficient at detecting DENV NS5 in the cell extracts, no specific binding between DENV NS5 and IGF2BP2-HA could be evidenced. Consistent with our different replication phenotypes between DENV and ZIKV, this strongly supports that the NS5/IGF2BP2 interaction is specific to ZIKV. The specificity of the IGF2BP2 interaction with ZIKV NS5 compared to DENV NS5 is discussed in the updated manuscript.

**Author response image 1. sa4fig1:** DENV NS5 is not specifically co-immunoprecipitated with IGF2BP2-HA in contrast to ZIKV NS5. Huh7.5 cells stably expressing IGF2BP2-HA (+) and control cells (-) were infected with ZIKV H/PF/2013 at a MOI of 10 or left uninfected. Two days later, cell extracts were prepared and subjected to RNase A treatment (+) or not (-) before anti-HA immunoprecipitations. The resulting complexes were analyzed by western blotting for their abundance in the indicated proteins.

**Reviewer #3 (Recommendations For The Authors):**
(1) Statistical analysis. Please clearly indicate what columns and error bars represent for bar graphs such as those presented in Figures 1A-D and F, Figures 2B-C, and bottom panels in DE, Figure 3, Figure 5B, Figure 6B-C, and Figures 9B-D and F. For instance, the mean of n independent experiments and standard deviation.

Information about the number of replicates, error bars, and statistical tests has been added for all figures in the legends.

(2) What is the scale in the Y-axis of Figure 2C? As shown, it is difficult to know what is the virus titer in knocked-down cells. Please use a linear scale or a log scale.

This is a linear scale of viral titers, which we have modified to make it clearer for the reader.

(3) Throughout the manuscript (e.g. Figures 1, 2, and 3) the fold reduction in titer is presented instead of the actual virus titers. I suggest showing the titer as it may be much more informative for the reader.

We prefer showing the data as fold reduction as they better reflect the IGF2BP2 knockdowninduced phenotypes across the independent biological replicates. Indeed, from one experiment to another, the reference titers in the control condition sometimes varies because of the cell passage or the lentiviral transduction efficiency for instance, especially when low multiplicities of infection are used. However, the reduction phenotype in foldchange observed upon IGF2BP2 knockdown was always consistent regardless of the titer value. Of note, all considered experiments had reference titers above 105 PFU/mL.

(4) Is it possible to perform a colocalization analysis of confocal images showing overlapping signals?

This has been done and the results of these analyses are included in the updated figures 4B, 5A, S4 and S5.

(5) Assessing the effect of Atlastin2 knockdown in virus yield and showing coimmunoprecipitation of Atlastin 2 with NS5 can add relevant information.

As mentioned in the discussion and above, ATL2 was already reported to be required for DENV and ZIKV replication in two independent studies (including one by one of the coauthors)(PMID: 31636417; 31534046). We have not tested whether ATL2 associates with NS5. However, new Fig. S7 of the revised manuscript shows that IGF2BP2/ATL2 is RNAdependent. This suggests that, as initially depicted in our model, IGF2BP2 associates with the ER (and thus, ATL2) after its binding to the viral RNA. Further interrogation into the role of atlastins in the flavivirus replication cycle is the focus of another ongoing IGF2BP2-unrelated study from one of the co-authors which will be reported elsewhere.